# A novel cis-regulatory element drives early expression of *Nkx3.2* in the gnathostome primary jaw joint

Jake Leyhr[1†], Laura Waldmann[1†], Beata Filipek-Górniok[2], Hanqing Zhang[3,4], Amin Allalou[3,4], Tatjana Haitina[1]*

[1]Subdepartment of Evolution and Development, Department of Organismal Biology, Uppsala University, Uppsala, Sweden; [2]Science for Life Laboratory Genome Engineering Zebrafish Facility, Department of Organismal Biology, Uppsala University, Uppsala, Sweden; [3]Division of Visual Information and Interaction, Department of Information Technology, Uppsala University, Uppsala, Sweden; [4]Science for Life Laboratory BioImage Informatics Facility, Uppsala, Sweden

**Abstract** The acquisition of movable jaws was a major event during vertebrate evolution. The role of NK3 homeobox 2 (Nkx3.2) transcription factor in patterning the primary jaw joint of gnathostomes (jawed vertebrates) is well known, however knowledge about its regulatory mechanism is lacking. In this study, we report a proximal enhancer element of *Nkx3.2* that is deeply conserved in most gnathostomes but undetectable in the jawless hagfish and lamprey. This enhancer is active in the developing jaw joint region of the zebrafish *Danio rerio*, and was thus designated as *jaw joint regulatory sequence 1* (JRS1). We further show that JRS1 enhancer sequences from a range of gnathostome species, including a chondrichthyan and mammals, have the same activity in the jaw joint as the native zebrafish enhancer, indicating a high degree of functional conservation despite the divergence of cartilaginous and bony fish lineages or the transition of the primary jaw joint into the middle ear of mammals. Finally, we show that deletion of JRS1 from the zebrafish genome using CRISPR/Cas9 results in a significant reduction of early gene expression of *nkx3.2* and leads to a transient jaw joint deformation and partial fusion. Emergence of this *Nkx3.2* enhancer in early gnathostomes may have contributed to the origin and shaping of the articulating surfaces of vertebrate jaws.

**\*For correspondence:**
tatjana.haitina@ebc.uu.se

[†]These authors contributed equally to this work

**Competing interest:** The authors declare that no competing interests exist.

## Editor's evaluation

In this elegant and important study, Leyhr et al. identify the first potent nkx3.2 jaw joint enhancer, which they show to be deeply conserved across gnathostomes and absent from jawless fishes. The data are convincing and beautifully presented, supporting the hypothesis that this enhancer arose with the origin of hinged jaws during vertebrate evolution and is required for some aspects of early joint development in zebrafish. The work has important implications both for our basic understanding of enhancer function and evolution as well as potential genetic causes of craniofacial defects in humans.

## Introduction

The establishment of jaw joints was one of the major events that enabled the evolutionary transition from jawless to jawed vertebrates. The earliest known articulated jaws are found in fossil placoderms from the late Silurian period 423 MYA (million years ago) (*Zhu et al., 2013*) The primary jaw joint of

non-mammalian gnathostomes, including actinopterygians, amphibians, reptiles, and birds is located within the first pharyngeal arch and articulates Meckel's cartilage and the palatoquadrate. These cartilages are derived from cranial neural crest cells that migrate into the first pharyngeal arch and later ossify into anguloarticular and quadrate bones, respectively (*Schilling and Kimmel, 1994*; *Tucker et al., 2004*).

The transition from jawless to jawed vertebrates and the underlying gene regulatory network changes are not yet fully understood. However, the transcription factor Nkx3.2 (Bapx1), which acts as a chondrocyte maturation inhibitor (*Provot et al., 2006*), is thought to have played a major role in the evolution of the primary jaw joint (*Cerny et al., 2010*). *Nkx3.2* displays focal expression in the first pharyngeal arch, between Meckel's cartilage/anguloarticular and the palatoquadrate/quadrate in non-mammalian vertebrates, as shown in zebrafish (*Miller et al., 2003*), *Xenopus* (*Square et al., 2015*), and python and chicken (*Anthwal et al., 2013*). In the lamprey, a jawless vertebrate, *Nkx3.2* is expressed in the ectomesenchyme surrounding the pharyngeal arches (*Miyashita, 2018*).

The importance of *Nkx3.2* for the development of the primary jaw joint has been shown in knockdown and knockout experiments carried out in zebrafish and *Xenopus*. Reduction or loss of *nkx3.2* expression led to absence of the joint and fusion of Meckel's cartilage and the palatoquadrate (*Lukas and Olsson, 2018b*; *Miller et al., 2003*; *Miyashita et al., 2020*; *Waldmann et al., 2021*). Overexpression of *nkx3.2* in *Xenopus* resulted in extra ectopic joints forming in the jaw cartilage (*Lukas and Olsson, 2018a*).

During the course of mammal evolution, the first pharyngeal arch elements and parts of the second arch underwent a morphological transition to form the middle ear ossicles in mammals (*Anthwal et al., 2013*; *Luo, 2007*). This transition was accompanied by the development of a secondary jaw joint articulating the squamosal and dentary. The middle ear consists of three major bones: the malleus, incus, and stapes, and the middle ear-associated bones: the gonium and tympanic ring which attach the ossicles to the skull. The incus and malleus are articulated by the incudomalleolar joint, homologous to the primary jaw joint of non-mammalian gnathostomes. *Nkx3.2* expression has been shown to be present in this joint, the tympanic ring, and the gonium, and mice homozygous for *Nkx3.2* knockouts display loss of the gonium and hypoplasia of the tympanic ring but minimal disruption to the incudomalleolar joint (*Tucker et al., 2004*). To what degree the gene regulatory network for these homologous joints is conserved is not fully understood.

In this study, we identified a novel gnathostome-specific cis-regulatory element, *jaw joint regulatory sequence 1* (JRS1), proximal to the *Nkx3.2* gene. We show that JRS1 has a highly conserved sequence and demonstrate that JRS1 sequences from multiple gnathostomes drive fluorescent reporter expression in the primary jaw joint of zebrafish, implying functional conservation between diverse clades. To test if JRS1 is essential for the jaw joint development, we generated a CRISPR/Cas9-induced enhancer deletion zebrafish line and show that homozygous mutants display transient dysmorphology and partial fusion of the jaw joint-articulating cartilages.

## Results

### *Nkx3.2* is located in a conserved syntenic region in vertebrates

In order to search for proximal conserved non-coding elements (CNEs), we first performed an analysis of the gene synteny around *Nkx3.2*, reasoning that if we find homologous genes upstream and downstream of *Nkx3.2* in different vertebrate species, the locus is unlikely to have undergone major rearrangements that would have disrupted the intergenic non-coding sequences. Our synteny analysis showed that in almost all examined jawed vertebrate genomes, *Nkx3.2* is located in a highly conserved syntenic region between *Bod1l1* (Biorientation of chromosomes in cell division protein 1-like 1) and *Rab28* (Ras-related protein Rab-28) genes (*Figure 1*). The upstream gene *Bod1l1* and downstream gene *Rab28* have the same orientation (located on the same DNA strand) as *Nkx3.2*. However, we could not identify *Bod1l1* in the coelacanth genome or *Rab28* in the zebrafish genome. On sea lamprey chromosome 11, *Nkx3.2* (LOC116941470, annotated as *fushi tarazu-like*) is located upstream from *Rab28* and next to *Nkx2.6* (LOC116941469), which is located on the opposite strand. *Bod1l1* (LOC116941243) is located more than 8.5 Mb upstream on chromosome 11 (*Figure 1*). BLAST searches identified an additional *Nkx3.2-like* sequence (LOC116940711) located on sea lamprey chromosome 9. However, this region did not share synteny with the *Nkx3.2*-containing genomic region. In

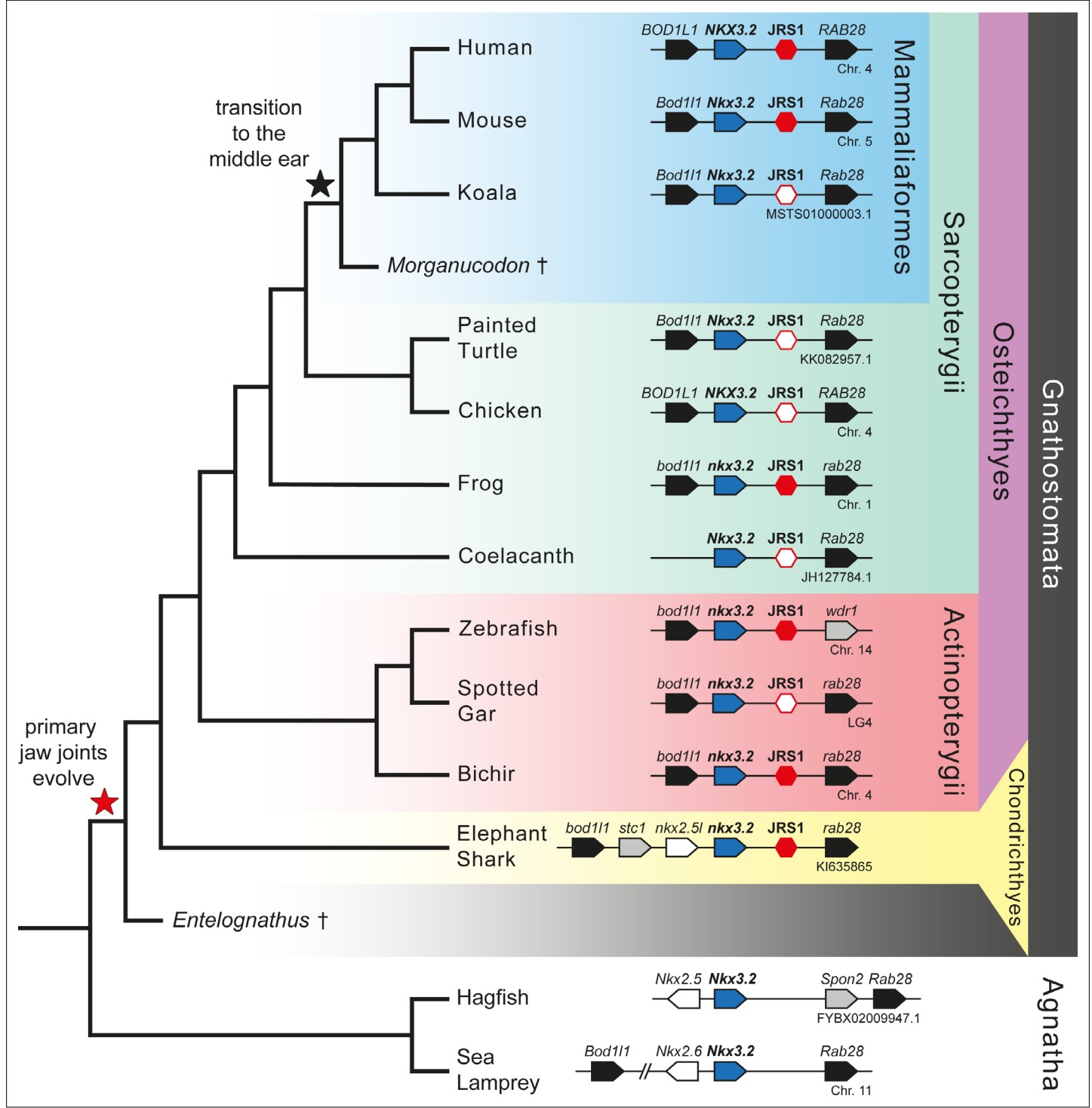

**Figure 1.** Gene synteny around *Nkx3.2* is well conserved in vertebrate genomes. Phylogenetic tree of vertebrates based on commonly accepted topology. Pointed boxes represent gene orientation with gene names indicated on top. Red hexagons mark the position of conserved non-coding element (CNE) (*jaw joint regulatory sequence 1*, JRS1) downstream of *Nkx3.2*, where filled hexagons mark CNEs selected for in vivo functional characterization in this study. The corresponding chromosome/contig number of each locus is indicated below the gene order schematic of each species. Daggers indicate extinct species.

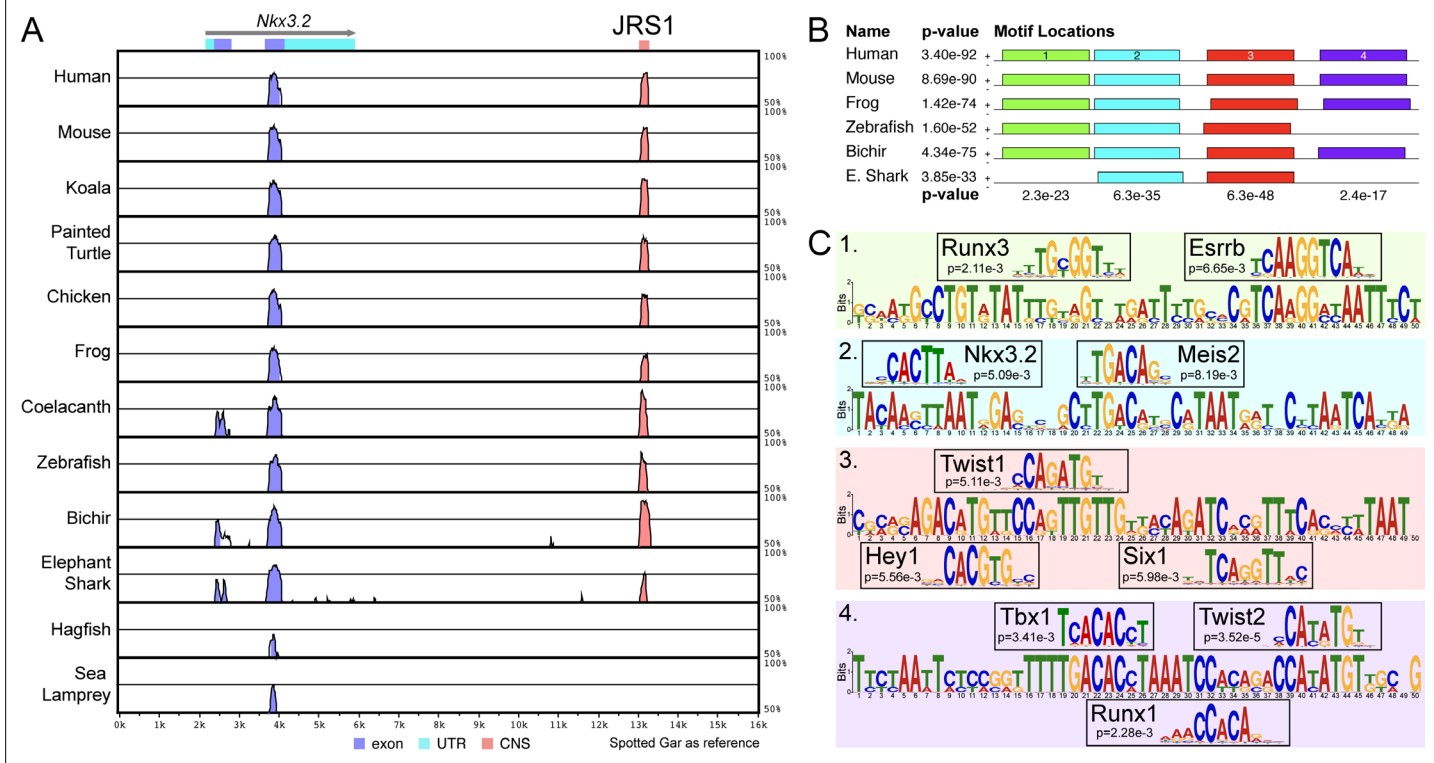

**Figure 2.** A conserved non-coding element, *jaw joint regulatory sequence 1* (JRS1), identified using mVISTA and MEME. (**A**) mVISTA alignment of vertebrate *Nkx3.2* loci, using the spotted gar locus as reference. Peaks indicate conserved sequences >50% identity, coloured peaks indicate >70% identity. Dark blue peaks indicate conserved exon sequences, pink indicates conserved non-coding sequences. (**B**) Shared sequence motifs (1–4) in the core ~245 bp sequence of JRS1 in different species identified with MEME analysis. p values indicated per species (horizontal) and per motif (vertical). (**C**) Relevant transcription factor-binding sites predicted by Tomtom at motifs 1–4 with associated p values for each match.

The online version of this article includes the following source data and figure supplement(s) for figure 2:

**Figure supplement 1.** Multiple sequence alignment, genome coordinates, and percent identity matrix of the *jaw joint regulatory sequence 1* (JRS1) conserved core.

**Figure supplement 2.** *Jaw joint regulatory sequence 1* (JRS1) is differentially bound by transcription factors in the first and second embryonic mouse branchial arches.

**Figure supplement 2—source data 1.** Genomic coordinates of transcription factor-binding enrichment at *jaw joint regulatory sequence 1* (JRS1) in E11.5 mouse branchial arch (BA) 1 and 2 determined by ChIP-seq assays.

the hagfish genome, *Nkx3.2* (ENSEBUG00000015739) is located on the contig FYBX02009947.1 next to *Nkx2.5* (ENSEBUG00000006582), but on the opposite DNA strand. *Rab28* (ENSEBUG00000007671) is located downstream from *Nkx3.2*, but there is an additional gene, annotated as Spondin 2 (*Spon2*, ENSEBUG00000016900) between them (*Figure 1*). BLAST searches also identified an *Nkx3.2-like* sequence on the genomic contig FYBX02010045.1 of hagfish, which was located in the first intron of *Sez6l* (ENSEBUG00000002694) but on the opposite strand. This region, however, also did not share synteny with the *Nkx3.2*-containing genomic region. These results indicated that the intergenic sequences flanking *Nkx3.2* were likely to be homologous in examined vertebrates, and therefore were appropriate to align in search of CNEs. In jawless sea lamprey and hagfish we searched for CNEs around both *Nkx3.2* and *Nkx3.2-like* sequences.

## A CNE is identified proximal to *Nkx3.2*

mVISTA analysis identified a conservation peak in the non-coding region downstream of *Nkx3.2*, between *Nkx3.2* and *Rab28*, for all examined gnathostome species: human, mouse, koala, painted turtle, chicken, frog, coelacanth, zebrafish, spotted gar, bichir, and elephant shark (*Figure 1*, *Figure 2A*). We could not identify the same conserved peak downstream of *Nkx3.2* in hagfish, either upstream or downstream of *Spon2*, in lamprey, or around the *Nkx3.2-like* sequences in these jawless species.

The peak sequences from several key gnathostome species (human, mouse, frog, zebrafish, bichir, and elephant shark) were extracted and a search for conserved motifs within these peak sequences was performed with MEME, identifying a core ~245 bp sequence (*Figure 2—figure supplement 1*) containing four conserved motifs, two of which (1 and 4) were absent in the elephant shark, and one of which (4) was absent in zebrafish (*Figure 2B*).

## JRS1 contains predicted transcription factor-binding sites for pharyngeal arch patterning

Comparison of each 49–50 bp motif within the JRS1 core sequence with known transcription factor-binding motifs revealed significant matches to several transcription factors known to be involved in pharyngeal arch patterning, skeletogenesis, and joint formation, consistent with a predicted regulatory role for JRS1 in the jaw joint expression of *Nkx3.2*. The predicted binding motifs are shown in *Figure 2C*; Runx3 and Esrrb in motif 1, Nkx3.2 and Meis2 in motif 2, Hey1, Twist1, and Six1 in motif 3, and Tbx1, Runx1, and Twist2 in motif 4.

Among these, the predicted Meis2-binding site in motif 2 (*Figure 2C*) is particularly notable as we were also able identify strong Meis enrichment precisely at the JRS1 locus in E11.5 mouse embryonic branchial arches (BAs) 1 (95.67-fold) and 2 (42.14-fold) in ChIP-seq datasets previously generated by *Donaldson et al., 2012* and *Amin et al., 2015* (*Figure 2—figure supplement 2*). In addition, in BA2 but not BA1, Meis binding is accompanied by Pbx (30.36-fold) and Hoxa2 (27.34) enrichment (*Figure 2—figure supplement 2*), indicating how differential activation of JRS1 may influence differential expression of *Nkx3.2* in the mandibular versus non-mandibular arches.

## Loss or divergence of JRS1 in most acanthopterygian fish

In addition to the broad sampling of gnathostome species, we performed a more detailed analysis of teleost fish species beyond just the zebrafish. These additional teleost species included the arowana (*Sclerophages formosus*), electric eel (*Electrophorus electricus*), Atlantic salmon (*Salmo salar*), Atlantic cod (*Gadus morhua*), slimehead (*Gephyroberyx darwinii*), pineconefsh (*Monocentris japonica*), soldierfish (*Myripristis murdjan*), alfonsino (*Beryx splendens*), cusk-eel (*Lucifuga dentata*), toadfish (*Thalassophryne amazonica*), mudskipper (*Periophthalmus magnuspinnatus*), seahorse (*Hippocampus comes*), tilapia (*Oreochromis niloticus*), amazon molly (*Poecilia formosa*), stickleback (*Gasterosteus aculeatus*), and pufferfish (*Tetraodon nigroviridis*).

Using a combination of mVISTA alignments and genome-wide BLASTN searches using the spotted gar and bichir as query sequences, we were able to identify JRS1 in all teleost species with the exception of most members of the clade Percomorpha: toadfish, mudskipper, soldierfish, seahorse, tilapia, amazon molly, stickleback, and pufferfish (*Figure 3*). The only percomorph species in which we could identify an element with both sequence and syntenic homology to JRS1 was the cusk-eel, a member of the earliest branching order of percomorphs: Ophidiiformes (*Ghezelayagh et al., 2022*; *Figure 3*). The sister order to the Percomorpha is Beryciformes, which includes the alfonsino and soldierfish. In both of these species, in addition to the Ophidiiform cusk-eel, BLASTN hits for JRS1 were much poorer than in other non-percomorph actinopterygians (*Figure 3*), but still identified a sequence fragment matching JRS1 in the intergenic region between *nkx3.2* and *rab28*. We classify these three species as possessing a 'JRS1 fragment'. Meanwhile, JRS1 in the pineconefish and slimehead (order Trachichthyiformes), and all non-acanthopterygian actinopterygians appears to be well conserved.

## Generation of *nkx3.2*(JRS1):mCherry transgenic lines

To test JRS1 for enhancer activity in vivo, we generated Tol2 reporter constructs with species-specific JRS1 sequences from human, mouse, frog, zebrafish, bichir, and elephant shark upstream of a membrane-tagged mCherry coding sequence. Embryos injected with JRS1 reporter constructs displayed mosaic reporter gene expression within the first pharyngeal arch elements Meckel's cartilage and palatoquadrate. At least two positive founders were identified for each injected construct and used for generation of stable *nkx3.2*(JRS1):mCherry transgenic lines, which allowed further characterization of enhancer activity.

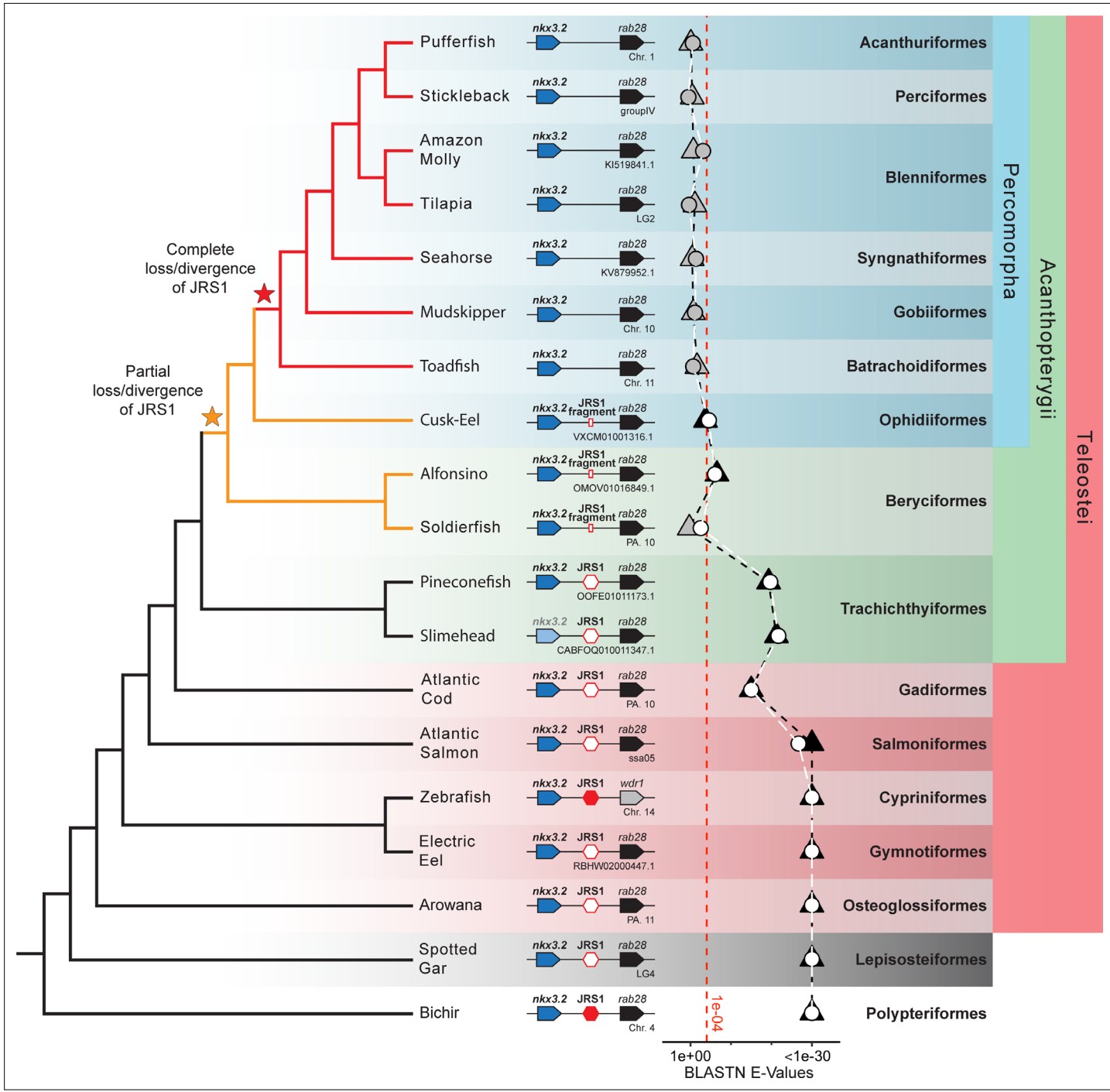

**Figure 3.** *Jaw joint regulatory sequence 1* (JRS1) is absent in most acanthopterygian fish. Phylogenetic tree based on *Hughes et al., 2018* and *Ghezelayagh et al., 2022*. Pointed boxes represent gene orientation with gene names indicated on top. The slimehead *nkx3.2* box is displayed as slightly grayed out as only the 3' UTR was present in the contig. Red hexagons mark the presence and position of JRS1, where filled hexagons mark enhancers selected for in vivo functional characterization in this study. Small red rectangles mark the position of JRS1 fragments. Below the gene order schematic of each species is marked the chromosome or contig containing this region. The plot of BLASTN *E*-Values represents the *E*-Values of the top hits to bichir (circles) and spotted gar (triangles) JRS1 query sequences in the genomes of each species. Shapes filled with white or black indicate that the top BLASTN hit using the bichir or spotted gar JRS1 query sequence, respectively, was found between the *nkx3.2* and *rab28* genes, while grey-filled shapes indicate the top hit was found in a different locus. The *x*-axis is truncated on the right side to a minimum of 1e−30 for ease of comparison, and the red dotted line indicates an *E*-Value of 1e−04. Taxonomic orders and higher clade classifications are shown on the far right.

The online version of this article includes the following source data for figure 3:

*Figure 3 continued on next page*

*Figure 3 continued*

**Source data 1.** The *E*-values of the top BLASTN hits using the bichir and spotted gar *jaw joint regulatory sequence 1* (JRS1) query sequences in the genomes of each species.

## Zebrafish JRS1 enhancer drives fluorescent reporter gene expression corresponding to endogenous *nkx3.2* expression

For analysing zebrafish JRS1 activity in the transgenic fluorescent reporter line, we performed live confocal imaging at different developmental time points. From 40 hours post fertilization (hpf) we detected persistent mCherry expression driven by the *nkx3.2* enhancer within the jaw joint-forming region of the first pharyngeal arch (*Figure 4*, *Video 1*). At 30 hpf the *nkx3.2*(JRS1):mCherry line

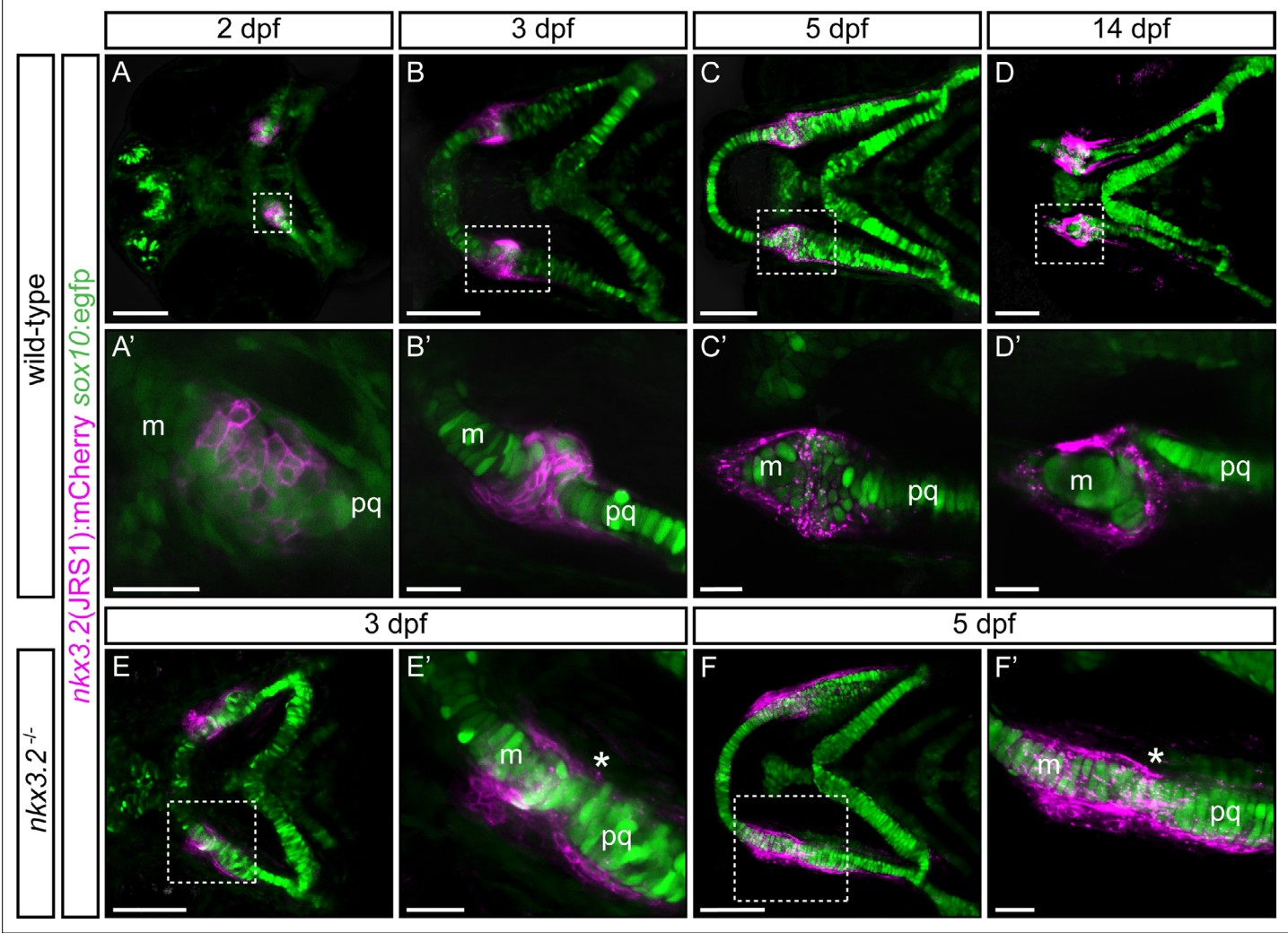

**Figure 4.** Zebrafish *jaw joint regulatory sequence 1* (JRS1) enhancer drives mCherry reporter gene expression in jaw joint-forming chondroprogenitor cells and partly overlaps with *sox10*:egfp expressing cells. (**A, A'**) At 2 dpf jaw joint progenitor cells express both GFP and mCherry. (**B, B'**) By 3 dpf cells start to differentiate and *nkx3.2*(JRS1):mCherry activity is restricted to jaw joint-forming interzone, overlapping with *sox10*:egfp. Single-labelled mCherry-expressing cells are surrounding the joint-forming region. (**C, C'**) At 5 dpf mCherry-expressing cells are restricted to the articulation forming area between Meckel's cartilage (m) and the palatoquadrate (pq). Double mCherry/GFP-expressing cells are restricted to posterior Meckel's cartilage and anterior palatoquadrate. (**D, D'**) At 14 dpf a clear joint cavity is visible. *nkx3.2*(JRS1):mCherry activity is restricted to the joint cavity and to both lateral and medial palatoquadrate. Dashed box in (**A–D**) is magnified in (**A'–D'**). (**A–D**) Represents maximum projection of confocal Z-stack, and (**A'–D'**) represents a single confocal image. In *nkx3.2⁻/⁻* mutants *nkx3.2*(JRS1):mCherry marks the cells outside of the fused jaw joint at 3 dpf (**E, E'**) and 5 dpf (**F, F'**). Dashed boxes in (**E, F**) are magnified in (**E', F'**). (**E, F**) and (**E', F'**) represent maximum projection of confocal Z-stack. Asterisks indicate the approximate location of the fusion site. Scale bars: 100 μm (**A–F**) and 25 μm (**A'–F'**).

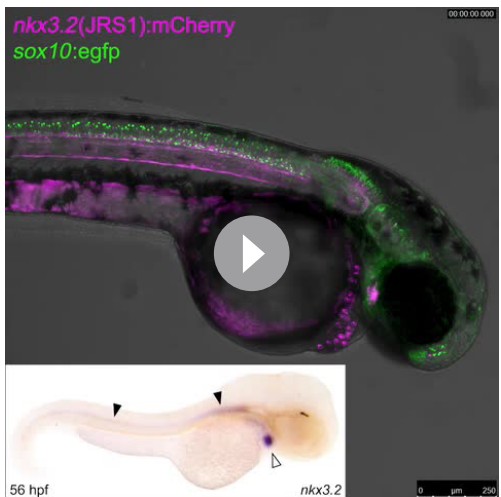

**Video 1.** *Jaw joint regulatory sequence 1* (JRS1) enhancer drives reporter expression in the early notochord and jaw joints. Lateral view of *nkx3.2*(JRS1):mCherry/*sox10:*egfp zebrafish embryo developing from 44 to 71 hpf. Inset in situ hybridization image of *nkx3.2* expression in a wild-type 56 hpf zebrafish embryo. White arrowhead indicates jaw joint expression domain while black arrowheads indicate expression in the notochord.

https://elifesciences.org/articles/75749/figures#video1

furthermore displayed mCherry-expressing cells in the notochord, which could be detected up to 3 days post fertilization (dpf) (***Video 1***).

*nkx3.2*(JRS1):mCherry/*sox10:*egfp double transgenic fish were used for the characterization of joint progenitor cells from 2 dpf. The *sox10:*egfp line labels neural crest cell-derived populations comprising pharyngeal arch cartilages (***Carney et al., 2006***). At the onset of chondrogenesis (2 dpf), confocal live imaging revealed overlapping expression of *nkx3.2*(JRS1):mCherry and *sox10:*egfp in condensed mesenchyme cells in the jaw joint-establishing zone (***Figure 4A, A′***). Single *sox10:*egfp expressing cells were located adjacent to this area, labelling first pharyngeal arch forming elements Meckel's cartilage and palatoquadrate (***Figure 4A′***). All labelled cells displayed rounded pentagonal morphology at this stage.

By 3 dpf *nkx3.2*(JRS1):mCherry-expressing cells were densely packed in the joint-forming region, overlapping with *sox10:*egfp expressing cells (***Figure 4B, B′***) and displayed a more elongated morphology. In Meckel's cartilage and the palatoquadrate, the majority of GFP-labelled chondrocytes underwent differentiation and maturation as indicated by the 'coin stack' arrangement (***Figure 4B, B′***). Double-labelled mCherry/GFP cells were furthermore present in the developing retroarticular process (RAP) (***Figure 4B, B′***). At both medial and lateral surfaces of the joint, single *nkx3.2*(JRS1):mCherry-expressing cells were lined up from the posterior Meckel's cartilage up to the anterior palatoquadrate.

mCherry expression became increasingly scattered in the membrane of labelled cells by 5 dpf. We observed a consistent presence of mCherry-positive cells on the lateral edge of the palatoquadrate and posterior Meckel's cartilage, partly overlapping with GFP-positive cells (***Figure 4C, C′***). The morphology of double-labelled cells in and surrounding the jaw joint-forming region was distinct from chondrogenic cells forming the main cartilage elements, by displaying smaller size and more rounded morphology (***Figure 4C, C′***). By 14 dpf, the joint cavity was evident and contained *nkx3.2*(JRS1):mCherry-expressing cells (***Figure 4D, D′***). mCherry-positive cells were maintained in the posterior part of Meckel's cartilage and we observed a line of mCherry-positive cells on both the lateral and medial sides of the palatoquadrate, reminiscent of the perichondrium, extending posteriorly towards the hyosymplectic of the second pharyngeal arch (***Figure 4D, D′***).

In order to characterize the *nkx3.2*(JRS1):mCherry-expressing cells in *nkx3.2* gene mutants we crossed previously reported *nkx3.2*[+/uu2803] mutant alleles (***Waldmann et al., 2021***) with *nkx3.2*(JRS1):mCherry/*sox10:*egfp fish and incrossed these heterozygotes to generate homozygous *nkx3.2*[uu2803/uu2803] mutants (abbreviated to *nkx3.2*[−/−]). Notably, *nkx3.2*(JRS1):mCherry-expressing cells were detected lining the outside of the cartilage in the fused jaw joint region of *nkx3.2*[−/−] mutants at 3–5 dpf (***Figure 4E, F and E′, F′***). The presence of double-labelled cells was not as apparent as in wild-type fish, however the scattering of the membranous mCherry signal was also noticeable in *nkx3.2*[−/−] mutants from 5 dpf (***Figure 4F, F′***).

## JRS1 activity is conserved in a range of gnathostome species

Generated transgenic lines containing JRS1 enhancer sequences from human, *Homo sapiens* (***Figure 5A***); mouse, *Mus musculus* (***Figure 5B***); frog, *Xenopus tropicalis* (***Figure 5C***); bichir, *Polypterus senegalus* (***Figure 5H***); and elephant shark, *Callorhinchus milii* (***Figure 5I***) displayed mCherry reporter

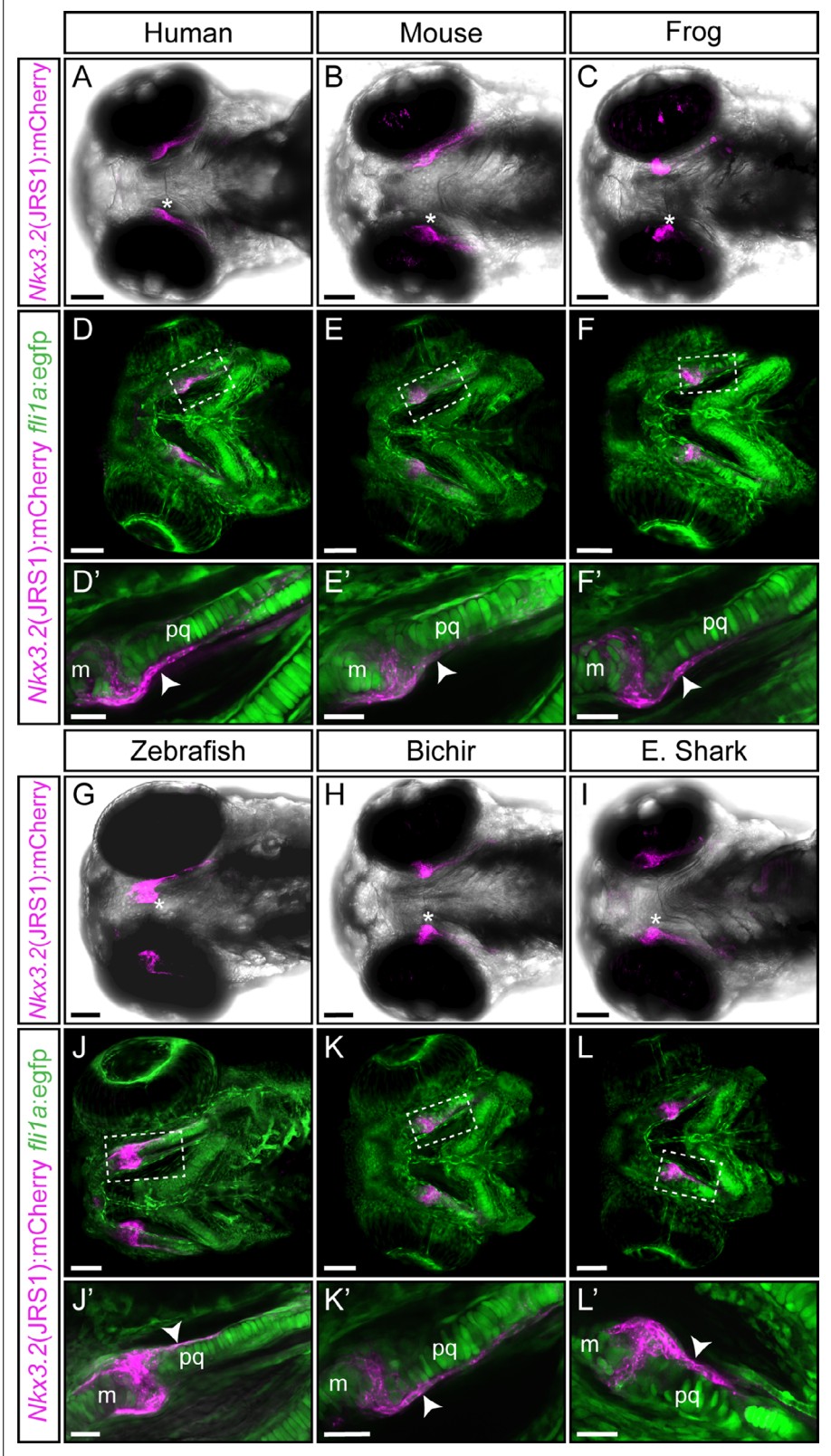

**Figure 5.** Functional conservation of the *jaw joint regulatory sequence 1* (JRS1) enhancer within tested gnathostome species. (**A–C, G–I**) Maximum projection images of 3 dpf transgenic zebrafish embryos (ventral view) driving mCherry expression in jaw joint and mandibular arch elements under the control of the JRS1 sequence of (**A**) human *Homo sapiens*, (**B**) mouse *Mus musculus*, (**C**) frog *Xenopus* tropicalis, (**G**) zebrafish *Danio rerio*, (**H**)

*Figure 5 continued on next page*

*Figure 5 continued*

bichir *Polypterus senegalus* and (**I**) elephant shark *Callorhinchus milii*. Asterisks mark a jaw joint. (**D–F** and **J–L**) Maximum projection images of 3 dpf nkx3.2(JRS1):mCherry transgenic zebrafish driving mCherry expression under the control of species-specific enhancer sequences with *fli1a*:egfp background reveals mCherry expression in GFP-labelled perichondrium cells. Dashed boxes are magnified in (**D'–F'**) and (**J'–L'**) as single confocal images. White arrowheads mark mCherry expression in the perichondrium. m: Meckel's cartilage; pq: palatoquadrate. Scale bars: 75 μm (**A–L**), 25 μm (**D'–F' and J'–L'**).

activity in the developing jaw joint (***Figure 5***), consistent with the zebrafish JRS1 activity (***Figure 5G***). In contrast to zebrafish JRS1 however, early (~30 hpf) mCherry expression in the notochord could not be detected in these lines. *fli1a:*egfp was used as background marker in order to investigate the potential JRS1 activity in the perichondrium. mCherry expression in perichondrium cells surrounding the developing jaw joint and palatoquadrate was present in all tested *nkx3.2*(JRS1):mCherry lines (***Figure 5D–F' and J–L'***).

## JRS1 deletion results in transient jaw joint dysmorphology and partial fusion

As JRS1 appears functionally conserved and active specifically in the jaw joint, we next investigated its function more closely by using CRISPR/Cas9 genome editing to generate a zebrafish line with the entire JRS1 enhancer sequence deleted from the genome. Two sgRNAs were targeted to sequences either side of JRS1 resulting in a 445 bp deletion spanning the conserved core of the JRS1 and ~100 bp flanking sequences (***Figure 6A***, ***Figure 6—figure supplement 1***). The resulting enhancer deletion allele is termed *nkx3.2*$^{\Delta JRS1}$. We generated homozygous mutant zebrafish (*nkx3.2*$^{\Delta JRS1/\Delta JRS1}$) and used Alcian blue and alizarin red staining to characterize the craniofacial morphology in comparison to heterozygotes and wild-type siblings at 5, 9, 14, and 30 dpf.

At all examined ages, the jaw joint morphology of wild-type and heterozygous mutants appeared indistinguishable (***Figure 6B, C, E, F, H, I***). At 5 dpf, homozygous *nkx3.2*$^{\Delta JRS1/\Delta JRS1}$ mutants tended to display a reduced RAP on Meckel's cartilage and the cartilage of the palatoquadrate often failed to form a pronounced convex joint process (***Figure 6D***). The result of this was a lack of a clearly defined concave–convex articulation in the jaw joint. Additionally, while no mutants displayed a complete fusion between the Meckel's and palatoquadrate cartilages, there was often a small number of chondrocytes (white arrowhead, ***Figure 6D***) connecting the two elements. Mouth gape angle did not appear to be affected, suggesting that the joint was at least partially flexible at this stage.

At 14 dpf, the RAP of homozygous mutants became more pronounced and comparable to wild-type and heterozygous siblings (***Figure 6G***). Meckel's cartilage and the palatoquadrate appeared to be fully separated, and the palatoquadrate joint process was more convex, resulting in a more concave–convex shape to the joint articulation. The RAP began to ossify as normal in all genotypes. By 30 dpf, the jaw joint of homozygous mutants was indistinguishable from wild-type and heterozygote siblings, with ossification of Meckel's cartilage, the RAP, and the palatoquadrate progressing normally (***Figure 6J***).

In situ hybridization analysis revealed a 44% (p = 0.0038) and 47% (p = 0.0002) reduction in *nkx3.2* expression in the jaw joints of *nkx3.2*$^{\Delta JRS1/\Delta JRS1}$ zebrafish mutants at 48 and 56 hpf, respectively, based on pixel intensity measurements (***Figure 6K–O***). The relative whole-body expression levels of *nkx3.2* quantified in JRS1 deletion mutants using qPCR detected no significant differences between the genotypes at 6 dpf (***Figure 6P***).

To quantify the subtle differences in posterior Meckel's cartilage shape in JRS1 mutant larvae, the heads of 9 dpf *nkx3.2*$^{+/+}$ (*N* = 12), *nkx3.2*$^{+/\Delta JRS1}$ (*N* = 10), and *nkx3.2*$^{\Delta JRS1/\Delta JRS1}$ (*N* = 8), were imaged using optical projection tomography (OPT), and the whole heads (***Figure 7—figure supplement 1***) and both jaw joints were rendered as maximum projections for a total of 60 jaw joints. The shape of Meckel's cartilage at the joint interface was analysed using 2D geometric morphometrics (***Figure 7A***), confirming the previously described observations that there was no significant difference between wild-type and heterozygous mutants (p = 0.20) while homozygous mutants differed significantly compared to them both (p = 0.0015). Analysis of morphospace also confirmed the tendency for homozygous mutants to display a reduced RAP resulting in a less concave surface interfacing with the palatoquadrate (***Figure 7A***). This phenotype is not fully penetrant, as there is some overlap between

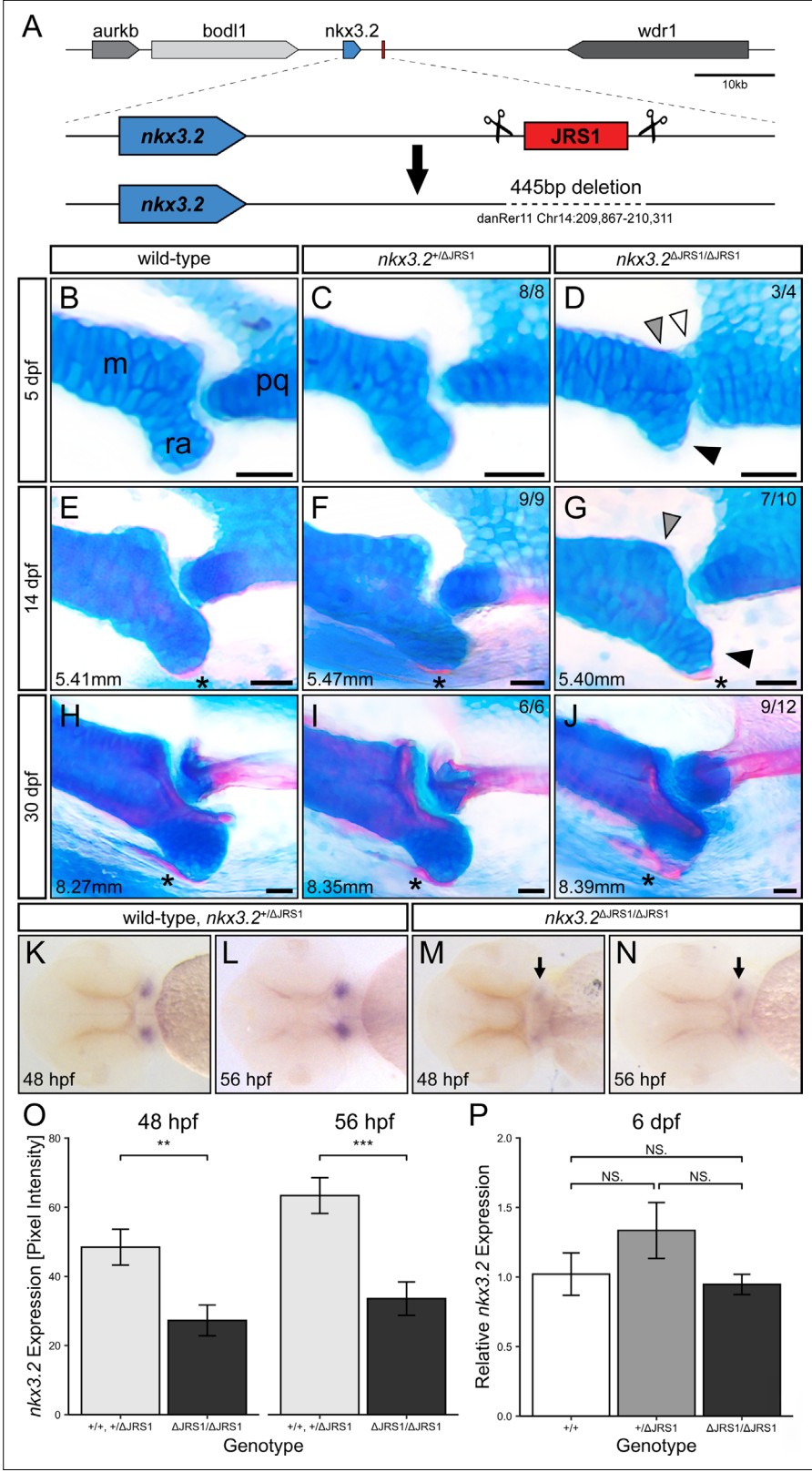

**Figure 6.** Homozygous *jaw joint regulatory sequence 1* (JRS1) enhancer deletion results in early jaw joint dysmorphology. (**A**) Schematic of the JRS1 deletion allele generation. (**B–J**) Alcian blue and alizarin red-stained jaw joints of representative wild-type, *nkx3.2*[+/ΔJRS1], and *nkx3.2*[ΔJRS1/ΔJRS1] zebrafish at 5, 14, and 30 dpf. Phenotypes are quantified in the top right of each panel. Standard lengths (mm) are given for 14 and 30 dpf juveniles. White

*Figure 6 continued on next page*

*Figure 6 continued*

arrowhead marks the partial fusion between Meckel's cartilage (m) and the palatoquadrate (pq). Grey arrowheads indicate the rounded, non-bulbous anteroposterior process of Meckel's cartilage. Black arrowheads mark the reduced retroarticular process (ra). Asterisks mark the ossifying retroarticular bone. Scale bars: 25 μm. (**K–N**) Representative embryos following in situ hybridization staining for *nkx3.2* in the ventral craniofacial region at 48 and 56 hpf. Arrows indicate the reduced expression in the jaw joint domains in *nkx3.2*$^{\Delta JRS1/\Delta JRS1}$ embryos. (**O**) *nkx3.2* expression levels at 48 and 56 hpf in wild-type and *nkx3.2*$^{+/\Delta JRS1}$ versus *nkx3.2*$^{\Delta JRS1/\Delta JRS1}$ zebrafish embryos, measured as pixel intensity of individual jaw joint expression domains in in situ stained embryos ($N = 16, 18, 18, 17$, respectively). (**P**) Relative whole-body *nkx3.2* expression levels at 6 dpf in wild-type, *nkx3.2*$^{+/\Delta JRS1}$, and *nkx3.2*$^{\Delta JRS1/\Delta JRS1}$ zebrafish, determined by qPCR. NS. denotes $p > 0.05$, ** $p < 0.01$, *** $p < 0.001$. Error bars represent mean ± standard error of the mean (SEM).

The online version of this article includes the following source data and figure supplement(s) for figure 6:

**Source data 1.** Relative pixel intensity values obtained from in situ images of *nkx3.2* expression in zebrafish at 48 and 56 hpf, and statistical analyses.

**Source data 2.** qPCR raw values.

**Figure supplement 1.** Alignment of wild-type and uu3731 *jaw joint regulatory sequence 1* (JRS1) deletion alleles with annotated MEME motifs, primers, and CRISPR gRNA targets.

---

the range of wild-type and mutant shapes, consistent with the variable severity of mutant joint phenotypes seen at 5 dpf. Averaged 3D renderings of the jaw joint at 9 dpf also displayed the partial fusion between Meckel's cartilage and the palatoquadrate previously described at 5 dpf (*Figure 7D, G*).

## Discussion

Previous studies have highlighted the importance of Nkx3.2 during primary jaw joint development and an overall role during chondrocyte maturation in various gnathostome species (*Miller et al., 2003*; *Provot et al., 2006*). In this study, we identified an *Nkx3.2* cis-regulatory element, JRS1, that is conserved in gnathostomes but absent from the homologous loci of jawless fishes, and investigated its activity in the jaw joint by generating transgenic reporter zebrafish lines and its function by generating a CRISPR/Cas9 deletion allele (*Figure 8*).

### JRS1 activity in the jaw joints

We combined gene synteny analysis and non-coding sequence conservation analysis using mVISTA and detected a conserved putative *nkx3.2* regulatory sequence (JRS1) in a number of gnathostome species. Next, we applied Tol2-mediated transgenesis to test JRS1 sequences from human, mouse, frog, zebrafish, bichir, and elephant shark for enhancer activity in zebrafish. The zebrafish *nkx3.2*(-JRS1):mCherry line labelled areas corresponding to the endogenous expression of *nkx3.2* in both the first pharyngeal arch and the early notochord (*Miller et al., 2003*; *Thisse and Thisse, 2005*). We detected mCherry-labelled cells in the developing jaw joint region, including the perichondrium proximal to the joint. *nkx3.2*(JRS1):mCherry expression overlapped with *sox10*:egfp-positive neural crest derived cells at early stages, consistent with the neural crest origin of jaw joint-establishing cells (*Carney et al., 2006*).

Notably, there was no apparent overlap of *nkx3.2*(JRS1):mCherry- and *sox10*:egfp-positive cells in *nkx3.2*$^{-/-}$ mutants that displayed the fused jaw cartilages as early as 3 dpf, in accordance with previous report by *Waldmann et al., 2021*. The majority of the *nkx3.2*(JRS1):mCherry-expressing cells lined the outside of the fused jaw joint cartilages, indicating that some joint or intermediate domain markers may still be present in the fusion region. This is reminiscent of the jaw joint fusion caused by the *dlx3b;4b;5a*-MO injection described by *Talbot et al., 2010*, where *trps1*, normally expressed in the articular cartilage of the wild-type jaw joint, was detected in cells surrounding the fusion in the morphants. Despite these cell patterning changes, *nkx3.2*(JRS1):mCherry-positive cells at 5 dpf displayed a similar accumulation of fluorescence reporter protein in the cell membrane of *nkx3.2*$^{-/-}$ mutants as in wild types, indicating the absence of cell identity changes.

The zebrafish jaw joint has previously been shown to display characteristics of synovial joints (*Askary et al., 2016*). Synovial joint development involves interzone formation after mesenchyme condensation. In contrast to adjacent, cartilage element-forming cells, interzone cells do not undergo chondrogenesis but become flattened and densely packed non-chondrogenic cells (*Craig et al.,*

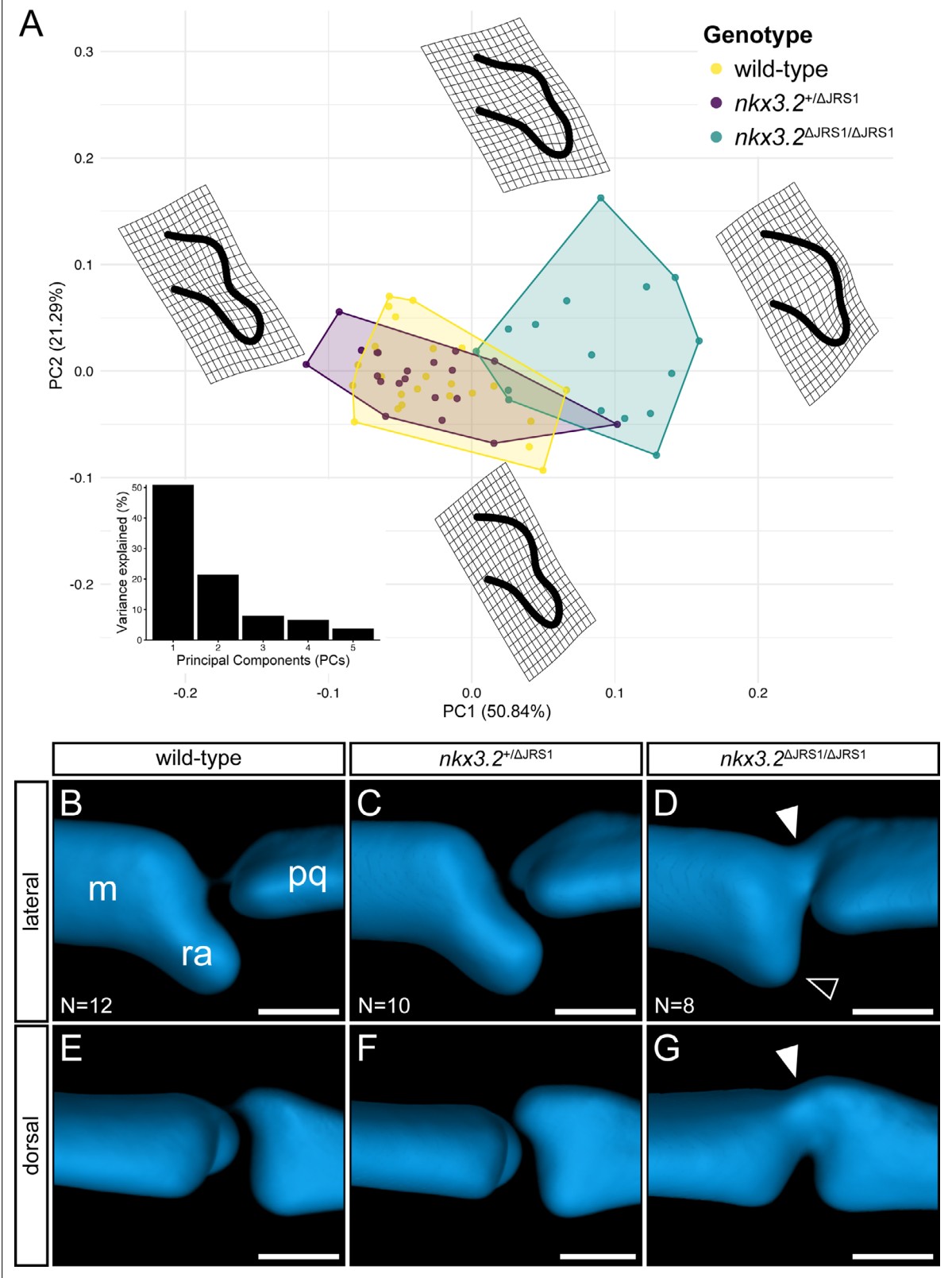

**Figure 7.** Geometric morphometric analysis of *jaw joint regulatory sequence 1* (JRS1) deletion phenotypes at 9 dpf. (**A**) Principal components analysis of geometric morphometric comparison of posterior Meckel's cartilage shape in wild-type, *nkx3.2*^+/ΔJRS1^, and *nkx3.2*^ΔJRS1/ΔJRS1^ zebrafish at 9 dpf. Thin-plate splines display the extremes of shape along PC1 and PC2. Inset is a histogram showing the percentage of variance explained by PCs 1–5. (**B–G**) Lateral and dorsal 3D renderings of the left jaw joint from averaged optical projection tomography (OPT) models of wild-type (*N* = 12), *nkx3.2*^+/ΔJRS1^ (*N* = 10),

*Figure 7 continued on next page*

*Figure 7 continued*

and *nkx3.2*<sup>ΔJRS1/ΔJRS1</sup> (*N* = 8) zebrafish at 9 dpf. White arrowheads mark the partial fusion between Meckel's cartilage (m) and the palatoquadrate (pq). Black arrowhead marks the reduced retroarticular process (ra). Scale bars: 25 μm.

The online version of this article includes the following source data and figure supplement(s) for figure 7:

**Source data 1.** Jaw joint landmarks TPS.

**Source data 2.** Jaw joint landmarks classifiers.

**Figure supplement 1.** Optical projection tomography of 9 dpf *jaw joint regulatory sequence 1* (JRS1) deletion zebrafish.

---

*1987*). *nkx3.2*(JRS1):mCherry expression at 3 dpf was found in cells displaying this characteristic inter-zone cell morphology. *nkx3.2*(JRS1):mCherry expression in the mandibular arch was consistent over time, displaying a strong fluorescence signal even at 14 dpf. The scattered membranous mCherry expression we could observe from 5 dpf onwards could be a consequence of the increasing extra-cellular matrix production by the interzone cells, necessary to facilitate the joint cavitation process (*Dowthwaite et al., 1998*; *Edwards et al., 1994*).

Perichondrium cell-derived signalling inhibits both chondrocyte proliferation and differentiation in mouse long bone development (*Alvarez et al., 2001*). Our finding of JRS1 activity in the perichon-drium cells lining Meckel's cartilage and the palatoquadrate suggests the expression of *nkx3.2* in perichondrium cells during chondrocyte maturation inhibition may be important for correct shaping and sizing of the cartilaginous elements. Whether and to what extent *nkx3.2* expression in the peri-chondrium regulates the shape of the mandibular arch and cartilage elements more broadly requires further investigation. With the JRS1 deletion experiments discussed below, we have started to provide some insight into this function.

## Transcription factor-binding motifs in JRS1

Functional testing of homologous JRS1 enhancer sequences from human, mouse, frog, bichir, and elephant shark resulted in species-specific mCherry expression almost identical to what was observed in the zebrafish *nkx3.2*(JRS1):mCherry transgenic line. These data suggest that the general land-scape of transcription factors binding to JRS1 is most probably conserved between gnathostomes, suggesting a highly conserved role for this cis-regulatory element. This is also supported by our prediction of transcription factor-binding motifs broadly shared between species that are associated with functions in the pharyngeal arch patterning and skeletogenesis (*Figure 2C*). This list of putative transcription factor-binding sites provides many candidates for validation and further research into the regulatory control of *Nkx3.2* in the pharyngeal arches.

Meis2 is known to function in the development of neural crest derivative tissues including cranial cartilage and bone (*Machon et al., 2015*), and fusions between Meckel's cartilage and the pala-toquadrate have been observed in *meis2*-knockdown zebrafish (*Melvin et al., 2013*). In addition, Meis proteins have been identified as cofactors for Hox transcription factor binding (*Mann et al., 2009*), along with Pbx proteins. Hox expression is absent from the mandibular arch but is important in establishing the identity of all the other pharyngeal arches. For example, *Hoxa2* expression estab-lishes the identity of second pharyngeal arch, as its absence results in the homeotic transformation of the second arch skeletal elements to a first arch identity (*Hunter and Prince, 2002*; *Trainor et al., 2002*), and the converse upon ectopic *Hoxa2* expression in the first or third arches: transformation into a second arch identity (*Grammatopoulos et al., 2000*; *Hunter and Prince, 2002*). The homeotic transformation of the second arch to a first arch identity in the absence of Hoxa2 in zebrafish includes the persistent expression of *nkx3.2* in the second arch (*Miller et al., 2004*), raising the possibility that Hoxa2 may act as a repressor of *nkx3.2* expression. The presence of a Meis2-binding site in motif 2, closely flanked by TAAT motifs favoured as Hoxa2-binding sites (*Berger et al., 2008*; *Donaldson et al., 2012*), are suggestive that such repressive activity may be mediated through JRS1. Indeed, an examination of E11.5 mouse branchial arch ChIP-seq datasets generated by *Donaldson et al., 2012* and *Amin et al., 2015* revealed that JRS1 is highly enriched in Meis binding sites in BA1 and highly enriched in overlapping Meis, Pbx, and Hoxa2 binding sites in BA2. This finding not only validates the presence of a Meis-binding site in JRS1 predicted using Tomtom, but also strongly suggests the presence of a Meis–Hoxa2–Pbx complex that may contribute to the repression of *nkx3.2* expression

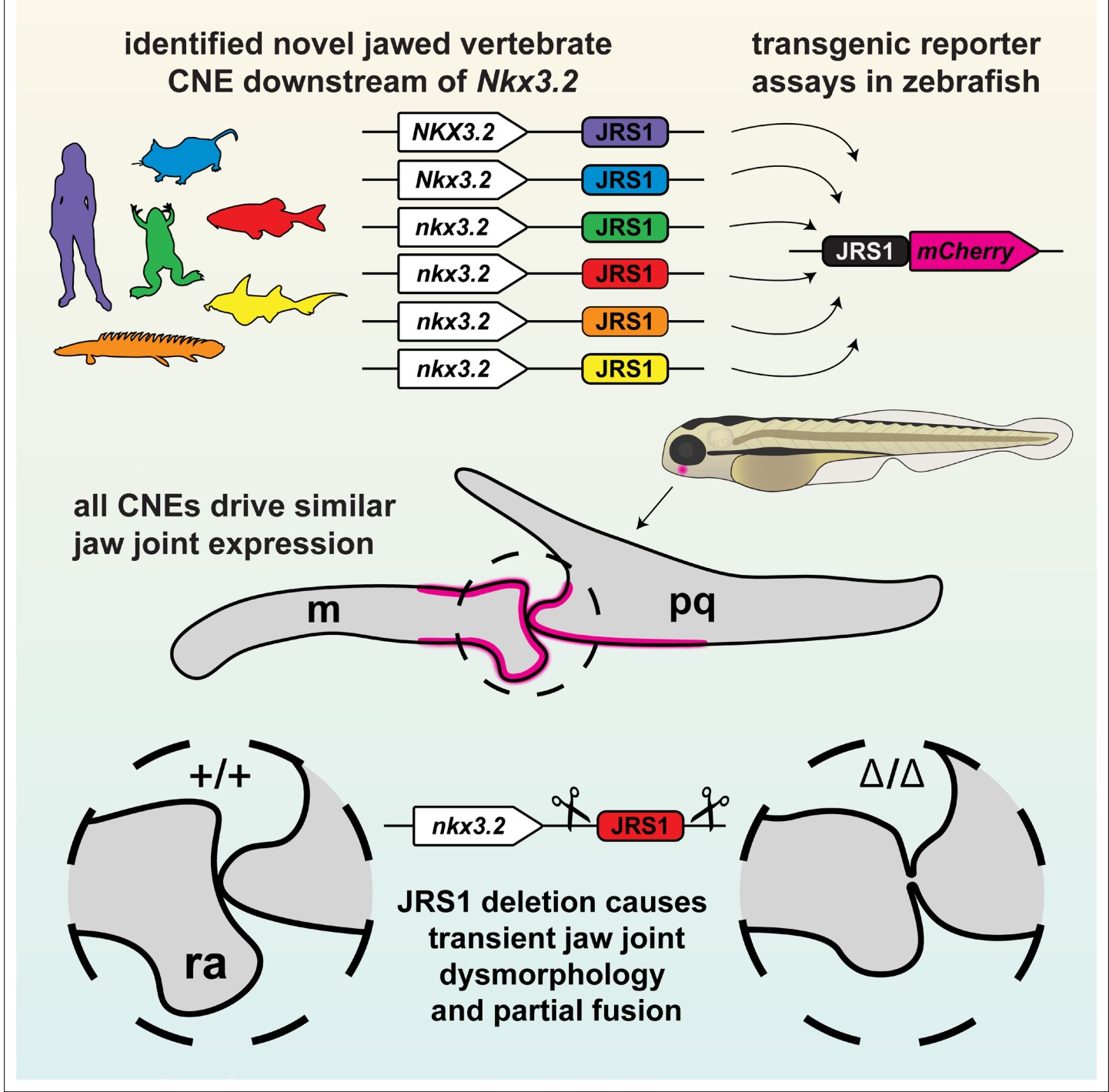

**Figure 8.** Graphical summary of this study. The conserved *jaw joint regulatory sequence 1* (JRS1) enhancer was first identified as a conserved non-coding element (CNE) *in silico*, then confirmed with transgenesis experiments of a range of species-specific JRS1 sequences driving fluorescent reporter expression in the jaw joint. Finally, the JRS1 enhancer was deleted from the zebrafish genome to reveal a transient jaw joint dysmorphology and partial fusion. m: Meckel's cartilage; pq: palatoquadrate; ra: retroarticular process.

in the second arch skeleton, either by causing JRS1 to act as a repressor of *nkx3.2* or by blocking activator activity.

Runx3 regulates target genes involved in chondrocyte development (*Wigner et al., 2013*), while Runx1 has been implicated in suppressing chondrocyte maturation by upregulating *Nkx3.2* expression (*Yano et al., 2019*). Hey1 is a transcriptional repressor expressed in the dorsal domain of the developing

pharyngeal arches downstream of Jagged-Notch signalling (*Zuniga et al., 2010*). The presence of a binding motif in JRS1 suggests that Hey1 may function downstream of Notch in repressing dorsal expression of *Nkx3.2*, as a dorsal expansion of *Nkx3.2* expression was observed when *Hey1* expression is reduced in response to defects in Jagged-Notch signalling (*Zuniga et al., 2010*). *Hey1* has also been found to be upregulated in osteoarthritis (OA) joint cartilage (*Chang et al., 2021*).

Tbx1 interacts with Hand2 and Edn1, key regulators of pharyngeal arch development, such that zebrafish *tbx1* mutants show drastic reductions in pharyngeal cartilage (*Piotrowski et al., 2003*). In addition to Tbx1, the same binding region matches many other Tbx factors as they all possess highly similar binding motifs, and some of these Tbx factors are also known to function in craniofacial development (*Papaioannou, 2014*). For example, *tbx22* expression mirrors that of *nkx3.2* in the developing zebrafish jaw joint (*Jezewski et al., 2009*; *Swartz et al., 2011*), and although the Tbx22-binding motif is absent from the JASPAR database, the conservation of Tbx factor-binding motifs suggests that Tbx22 can likely bind at this motif in JRS1, implying a possible role for Tbx22 in jaw joint formation upstream of Nkx3.2. Twist1 and Twist2 have been shown to function in both promoting and inhibiting chondrogenesis in different contexts (*Cleary et al., 2017*; *Reinhold et al., 2006*; *Takai et al., 2019*), and are expressed in the zebrafish craniofacial skeleton (*Germanguz and Gitelman, 2012*). *TWIST1* is upregulated in OA (*Hasei et al., 2017*) while *NKX3.2* is downregulated (*Caron et al., 2015*; *Oh et al., 2021*), suggesting that repression of *NKX3.2* expression by TWIST1 may contribute to OA articular cartilage pathology. *SIX1* has been found to have expression restricted to articular cartilage in porcine knee joints (*Hissnauer et al., 2010*), and has a role in craniofacial skeletogenesis promoting Jagged-Notch signalling and repressing *Edn1* expression in the dorsal pharyngeal arches (*Tavares et al., 2017*). It is also upregulated by the aforementioned Tbx1 (*Guo et al., 2011*).

A potential function of Esrrb in skeletogenesis has not been described, but closely related members of the estrogen-related receptor family with similar binding motifs, ESRRA and ESRRG, are known to function in promoting chondrogenesis and cartilage degradation in OA (*Kim et al., 2015*; *Son et al., 2017*; *Tang et al., 2021*). Finally, the presence of an Nkx3.2-binding motif in JRS1 suggests a potential for autoregulation of *nkx3.2* expression, establishing a steady expression level at a reduced metabolic cost (*McAdams and Arkin, 1997*). However, the significance of this potential autoregulation may be minor or restricted in time, as we did not observe noticeable reductions in JRS1 reporter activity in *nkx3.2*$^{-/-}$ gene mutants (*Figure 4E, F*).

## Evolution of JRS1 in vertebrates

The localization of *Nkx3.2* to the intermediate domain of the first pharyngeal arch has been suggested as one of the key drivers of the evolution of vertebrate jaws (*Cerny et al., 2010*). As such, the identification of JRS1 as a jaw joint-specific enhancer of *Nkx3.2* that is conserved in gnathostomes but absent from the jawless hagfish and lamprey suggests that JRS1 evolved in the gnathostome stem group after the split with Agnatha, and may have played a key role in the early evolution of jaws. It is also noteworthy that JRS1 appears to be functionally conserved in mammals as the primary jaw joint region, along with its *Nkx3.2* expression domain, has moved to the middle ear to form the joint between the malleus and incus (*Anthwal et al., 2013*; *Luo, 2007*). This suggests that JRS1 may have continued to function in driving *Nkx3.2* expression in the mammalian incudomalleolar joint despite the significant morphological changes relative to the ancestral jaw joint.

The elephant shark JRS1 sequence drove reporter expression in the joint of the zebrafish first (mandibular) pharyngeal arch, consistent with other gnathostomes. In the related elasmobranch chondrichthyans, gene expression of *nkx3.2* has been reported in the intermediate domains of all pharyngeal arches, including the hyoid and gill arches (*Compagnucci et al., 2013*; *Hirschberger et al., 2021*). If we assume elephant shark *nkx3.2* gene expression mirrors that of elasmobranchs and that JRS1 drives *nkx3.2* gene expression in all pharyngeal arches in chondrichthyans, our results would support the conclusion that much of the gene regulatory landscape of osteichthyan mandibular joints is found more broadly in the pharyngeal arches of chondrichthyans (*Hirschberger et al., 2021*). On the other hand, the presence of a conserved Meis-binding site and TAAT core Hox-binding motifs in elephant shark JRS1 may indicate that JRS1 could be differentially activated according to the nested Hox expression in different pharyngeal arches (*Oulion et al., 2011*). In this case, other enhancers would likely contribute to post-mandibular arch expression of *nkx3.2* in chondrichthyans.

JRS1 appears to be absent in all analysed percomorph (PERC) teleost fish with the exception of the cusk-eel, and only a small conserved sequence fragment is present in the cusk-eel, alfonsino, and soldierfish – acanthopterygian fish from the orders Ophidiiformes (O) and Beryciformes (B) that narrowly post- and pre-date the phylogenetic origin of percomorph fish, respectively (*Ghezelayagh et al., 2022*). This suggests that while JRS1 may have been lost in the last common ancestor of all percomorphs with the exception of ophidiiforms (PERC-O), a process of loss or adaptive sequence divergence may have begun earlier, in a common ancestor of both beryciforms and percomorphs (PERC+B). The unambiguous presence of JRS1 in the pineconefish and slimehead (order Trachichthy-iformes) rules out the possibility that JRS1 was lost in the common ancestor of all acanthopterygians. Nevertheless, it is tempting to speculate whether the apparent absence of JRS1 could be related to the evolution of the characteristic upper jaw protrusability found in many acanthopterygians (*Lauder and Liem, 1969*; *Motta, 1984*), as this altered functional morphology may have led to changes in selective pressures acting on the mandibular jaw joint.

Alternatively, JRS1 may have undergone rapid sequence evolution in PERC+B relative to other tele-osts, and is simply too diverged to be detected based on sequence homology (*Lee et al., 2011*; *Ravi and Venkatesh, 2018*). However, when aligning only PERC+B species *nkx3.2* loci, we were unable to find a JRS1-like conserved sequence within this clade, which is more consistent with a total deletion or degeneration of JRS1 in the last common ancestor of PERC+B. We cannot rule out the presence of novel or pre-existing shadow enhancers in this clade that may have taken over the role of JRS1 in regulating *nkx3.2*, as our search for conserved sequences was limited to the *bod1l1-nkx3.2-rab28* locus while other enhancers may be present in more distant intergenic regions (*Hong et al., 2008*).

## Jaw joint morphology and *nkx3.2* expression levels associated with JRS1 deletion

Experimental JRS1 deletion did not phenocopy the striking jaw joint fusion seen in zebrafish *nkx3.2* gene knockout mutants (*Miyashita et al., 2020*; *Waldmann et al., 2021*), suggesting the deletion does not abolish *nkx3.2* expression at the time points analysed. However, as subtle jaw joint pheno-types were still evident in homozygous mutants, most notably the reduction of the RAP and the partial fusion observed at 5–9 dpf, the loss of JRS1 must have some local effect on *nkx3*.2 expression, consis-tent with the specific jaw joint activity seen in transgenic reporter fish. Indeed, by quantifying the jaw joint expression of *nkx3.2* in zebrafish embryos by measuring in situ hybridization staining intensity, we were able to measure a ~45% decrease at 48–56 hpf in homozygous JRS1 deletion mutants relative to wild types and heterozygous mutants. *Castellanos and Quintana, 2021* reported mild jaw joint phenotypes reminiscent of those seen in our homozygous JRS1 mutants in 5 dpf *hspg2* morphants that were associated with a ~30% reduction in *nkx3.2* gene expression measured at 4 dpf. Their results and ours are consistent with earlier work indicating that a reduction in early *nkx3.2* expression levels can result in more subtle jaw joint phenotypes, increasing in severity with increasing morpholino dosage (*Miller et al., 2003*).

The jaw joint phenotype in homozygous JRS1 mutants appears to be rescued by approximately 14 dpf, suggesting that either *nkx3.2* expression recovers after the initial decrease caused by the absence of JRS1, or that late *nkx3.2* expression is less important for shaping the developing jaw joint and other factors take over instead. At 6 dpf, we could not detect any significant differences in whole-body *nkx3.2* expression levels, supporting the former hypothesis. If this is the case, it suggests that it takes approximately 1 week for the phenotype to be rescued once normal *nkx3.2* expression is restored. It also indicates that JRS1 function is most important to the early jaw joint expression of *nkx3.2*, prior to 6 dpf, and may contribute relatively less to later expression. Alternatively, it is possible that *nkx3.2* expression levels in the jaw joint do not recover, and the absence of significant differences observed at 6 dpf is the result of the reduced jaw joint expression signal being masked by expression elsewhere in the body.

The results of single enhancer deletions in previous studies vary widely from no phenotypic effect at all (*Cunningham et al., 2018*; *Osterwalder et al., 2018*), to subtle effects (*Dickel et al., 2018*), to strong effects approaching or phenocopying gene knockouts (*Dobrzycki et al., 2020a*; *Sagai et al., 2005*). More severe phenotypes relative to homozygous gene knockouts likely scale with greater reductions in gene expression (*Osterwalder et al., 2018*), although in some cases the phenotypic effects may instead rely on an expression threshold being breached (*Lam et al., 2015*). It is common

for developmental genes to be regulated by multiple enhancers, contributing to expression in a range of tissues and conferring a degree of functional redundancy (*Chen et al., 2016*; *Hobert, 2010*; *Osterwalder et al., 2018*; *Wang and Goldstein, 2020*). In this light, it is perhaps unsurprising that deletion of JRS1 does not completely abolish *nkx3.2* gene expression in the jaw joint, as there are likely other as-yet undiscovered enhancers that also contribute to *nkx3.2* regulation in this location in zebrafish and likely other gnathostome species as well. This work highlights the importance of screening enhancer deletion mutants for phenotypic effects at multiple developmental stages, as the dynamic nature of gene expression by temporally specific enhancers can mask or otherwise rescue phenotypes observed earlier in development. We encourage future studies to assess enhancer activity and gene expression to quantify these temporal dynamics, especially in known multi-enhancer systems.

## Conclusion

We report the identification and functional characterization of a novel *Nkx3.2* enhancer, JRS1, with activity specific to the developing primary jaw joint that contributes to early *nkx3.2* gene expression. As this enhancer is conserved in sequence and activity in gnathostomes including both bony and cartilaginous fish, we conclude that it arose early in gnathostome evolution and may have been one of the earliest novel cis-regulatory elements to facilitate the evolution of jaws from the ancestral jawless state by driving the localization of *Nkx3.2* expression into the jaw joint and contributing to the early jaw joint morphology. Secondary loss of JRS1 in most acanthopterygians may be related to the evolution of novel jaw morphologies in this group. Further study of JRS1 and other as-yet undiscovered *Nkx3.2* enhancers will provide new insights into developmental regulatory network responsible for the evolution and diversification of gnathostome jaws.

# Materials and methods

## Conserved synteny analysis

The synteny analysis was performed on the genomic regions containing the *Nkx3.2* gene (alternative names *Nkx-2* and *Bapx1*). Synteny data including upstream and downstream genes from *Nkx3.2* were extracted from the Ensembl and NCBI databases for several genomes: human, *Homo sapiens* (GRCh38.p12); mouse, *Mus musculus* (GRCm38.p6); koala, *Phascolarctos cinereus* (phaCin_unsw_v4.1); painted turtle, *Chrysemys picta* (Chrysemys_picta_bellii-3.0.3); chicken, *Gallus gallus* (GRCg6a); tropical clawed frog, *Xenopus tropicalis* (*Xenopus*_tropicalis_v9.1); coelacanth, *Latimeria chalumnae* (LatCha1); zebrafish, *Danio rerio* (GRCz11); spotted gar, *Lepisosteus oculatus* (LepOcu1); elephant shark, *Callorhinchus milii* (Callorhinchus_milii-6.1.3); inshore hagfish, *Eptatretus burgeri* (Eburgeri_3.2); and sea lamprey, *Petromyzon marinus* (kPetMar1). *Nkx3.2* sequences were identified in hagfish and lamprey genomes with TBLASTN searches using the protein sequence of the spotted gar Nkx3.2 (ENSLOCG00000009892) as a query. A genomic scaffold containing the *Nkx3.2* locus was obtained from genomic databases of the bichir, *Polypterus senegalus* (*Mashima et al., 2016*; *Tatsumi et al., 2016*), now publicly available as ASM1683550v1.

## Conserved non-coding sequence identification and motif analysis

The genomic region surrounding the *Nkx3.2* gene, between the two immediately flanking genes (typically *Bod1l1* and *Rab28*) were collected for a number of vertebrate species and submitted to mVISTA (*Frazer et al., 2004*) using the default settings to search for conserved non-coding sequences. Non-coding sequences representing peaks of conservation were collected for human, mouse, frog, zebrafish, bichir, and elephant shark and further analysed with the MEME Suite (*Bailey et al., 2009*) to distinguish the conserved core of the region identified with mVISTA. Additional sequence alignment performed using Clustal Omega (*Madeira et al., 2019*) and manually curated. Conserved sequence motifs in the core sequences were discovered with MEME (*Bailey and Elkan, 1994*) using classic discovery mode and searching for four motifs with zero or one occurrence per sequence. Each of the discovered statistically significant motifs was matched against transcription factor-binding motifs in the JASPAR CORE vertebrates database using Tomtom (*Gupta et al., 2007*).

To search for JRS1 in a range of teleost species, additional sequences were extracted from the Ensembl and NCBI Genome databases, representing the *nkx3.2* gene sequence and the downstream non-coding sequence until the start of the *rab28* gene. These teleost species included the

arowana, *Scleropages formosus* (fSclFor1.1); electric eel, *Electrophorus electricus* (Ee_SOAP_WITH_SSPACE); Atlantic salmon, *Salmo salar* (ICSASG_v2); Atlantic cod, *Gadus morhua* (gadMor3.0); Darwin's slimehead, *Gephyroberyx darwinii* (GCA_900660455.1); pineconefish, *Monocentris japonica* (ASM90032336v1); pinecone soldierfish, *Myripristis murdjan* (fMyrMur1.1); splendid alfonsino, *Beryx splendens* (ASM90031256v1); toothed Cuban cusk-eel, *Lucifuga dentata* (Ldentata1.0); Amazon toadfish, *Thalassophryne amazonica* (fThaAma1.1); Korean mudskipper, *Periophthalmus* magnuspinnatus (fPerMag.1.pri); tiger-tail seahorse, *Hippocampus comes* (H_comes_QL1_v1); Nile tilapia, *Oreochromis niloticus* (O_niloticus_UMD_NMBU); Amazon molly, *Poecilia formosa* (Poecilia_formosa-5.1.2); three-spined stickleback, *Gasterosteus aculeatus* (BROAD S1); and green spotted pufferfish, *Tetraodon nigroviridis* (TETRAODON 8.0). For Darwin's slimehead, the *nkx3.2* coding sequence was not included in the relevant contig, as it terminated at the 5′ boundary of the 3′ UTR. mVISTA analysis was performed as described above, and additional BLASTN searches were performed with the following parameters: match/mismatch scores: 1, −1; gap costs: 3, 2; seed word size: 11.

## Construct cloning

Genomic DNA *from* human *H. sapiens*, mouse *M. musculus*, frog *X. tropicalis*, zebrafish *D. rerio*, bichir *P. senegalus*, and elephant shark *C. milii* was used to amplify identified JRS1 sequences. Forward primers contained four guanine residues at the 5′ end followed by attB4 (5′-ACAACTTTGTATAGAA AAGTT-3′) attachment sites and finally the species-specific template sequence: human 5′-GTCAC ACAGCTTGGAATTGGTG-3′, mouse 5′-AGTTTTACAGGTTCCTAGCCCATAC-3′, frog 5′-TCTGAACTG TTTTGCCCACATT-3′, zebrafish 5′-AGACGTGATGCTGTGACACGCTAACTGCTG-3′, bichir 5′-GAACC GAGTGCTTTACAATTAGGTA-3′, elephant shark 5′-GAATGGAGTCACACGATAGTAATCC-3′. Reverse primers contained four guanine residues at the 5′ end, attB1r (5′-ACTGCTTTTTTGTACAAACTTG-3′) attachment sites, and the adenovirus E1b minimal promoter sequence followed by species-specific template sequences: human 5′- AAGTGGTTCAAAGGCTAAAGTT-3′, mouse 5′-CCTCATTGCTCCA CCTCTCT-3′, frog 5′-ACATTGGCACTGACAGGTAAAC-3′, zebrafish 5′-GATTTACATTTTGACGTCAAT -3′, bichir 5′- TTTCGAAATATTTGATACCGACAGT-3′, elephant shark 5′-AAAGTGCATTGTGAACAAATA GACA-3′. PCR products were subsequently recombined into pDONR P4-P1R donor vectors by the BP reaction according to the MultiSite Gateway cloning protocol (*Kwan et al., 2007*; Invitrogen). For generating expression clones, entry clones created by BP reaction were used as 5′ elements, pME-mCherry-CAAX as the middle entry clone, p3E-polyA as the 3′ entry clone and pDestTol2pA2 as the destination vector (*Kwan et al., 2007*). The LR reaction was performed according to the MultiSite Gateway cloning protocol (Invitrogen).

## Zebrafish transgenic lines

One-cell stage zebrafish (*D. rerio*) embryos were microinjected with freshly mixed injection solution containing the expression clone plasmid (175 ng/µl) and transposase mRNA (125 ng/µl) according to *Fisher et al., 2006*. Injected embryos were screened at 3 dpf for mosaic *nkx3.2*(JRS1):mCherry expression. F0 embryos displaying the strongest mosaic *nkx3.2*(JRS1):mCherry expression were selected and raised to sexual maturity (90 dpf). Mature adults were outcrossed with AB fish. Positive Tg(*nkx3.2*(JRS1):mCherry) F1 embryos were raised to establish stable transgenic lines. For generating double transgenic zebrafish, the Tg(*nkx3.2*(JRS1):mCherry) line was crossed with previously published zebrafish lines Tg(*fli1a*:egfp) (*Lawson and Weinstein, 2002*) or Tg(*sox10*:egfp) (*Carney et al., 2006*).

For comparison between wild-type and homozygous *nkx3.2* gene mutant zebrafish, the previously reported *uu2803* null allele (*Waldmann et al., 2021*) was crossed with double transgenic line *nkx3.2*(-JRS1):mCherry/*sox10*:egfp and further incrossed to produce *nkx3.2*$^{-/-}$ fish.

## Confocal live-imaging microscopy

Confocal microscopy was performed on an inverted Leica TCS SP5 microscope using Leica Microsystem LAS-AF software. Embryos were sedated with 0.16% MS-222 and embedded in 0.8% low-melting agarose on the glass bottom of the 35 mm dishes. To prevent drying, embedded embryos were covered with system water containing 0.16% MS-222. Images are presented as single images or maximum projections, as specified.

## Enhancer deletion using CRISPR/Cas9

Two gRNAs targeting flanking regions of the zebrafish *nkx3.2* JRS1 enhancer were selected with the use of CRISPOR online design tool (*Concordet and Haeussler, 2018*): 5′-TGACGAGAGGAGCGACA CGC-3′ and 5′-GCGTGTCGCTCCTCTCGTCA-3′. The gRNAs were prepared as previously described (*Varshney et al., 2015*). In short, annealing of the two oligos containing T7 promoter, target-specific sequence, where the first two nucleotides were modified for the T7 synthesis needs, and the guide core sequence was performed. Reaction products were used as a template for the in vitro transcription (HiScribe T7 High Yield RNA Synthesis Kit, England Biolabs) and purified. Cas9 mRNA was prepared from the p-T3Ts-nCAs9 plasmid (46757 Addgene) including the restriction enzyme digestion (XbaI, New England Biolabs), in vitro transcription (mMESSAGE mMACHINE T3 Transcription Kit, Life Technologies) and product purification. Fertilized eggs were obtained by natural spawning of AB zebrafish and injected at the one-cell stage with 150 pg of *Cas9* mRNA and 50 pg of each sgRNA in RNase-free H$_2$O. The efficiency of the targets was estimated by the CRISPR-STAT method (*Carrington et al., 2015*). Sequences of the primers used for activity testing and genotyping were 5′-GTGACACGCTAAC TGCTGGA-3′ and 5′-GAACATCCTTCATGGGCTTC-3′. All primer and gRNA sequences are shown schematically in *Figure 6—figure supplement 1*.

Injected F0 fish were raised to adulthood and individually outcrossed with AB zebrafish. Clutches of 5 dpf larvae were genotyped to determine the presence of the enhancer deletion in the germline of F0 parents. Three batches of 8–12 randomly selected larvae were sacrificed and lysed in a solution of 150 µl 50 µM NaOH for 20 min at 95°C. The lysis solution was subsequently stabilized by adding 100 µl 0.1 mM Tris. 1 µl of the resulting lysis solution was added into a 25 µl PCR reaction containing Platinum Taq DNA Polymerase and the two primers flanking the JRS1 deletion site, and was run for 35 cycles. Resulting PCR products were run on 1% agarose gels to determine their size. The PCR product representing the JRS1 deletion allele was 352 bp in length, and easily distinguishable from 790 bp wild-type allele. Four deletion-positive F0 zebrafish were outcrossed with Tg(*fli1a*:GFP) fish (*Lawson and Weinstein, 2002*) to produce four F1 lines, which were raised to adulthood and genotyped by fin clip. One of the four heterozygous lines (deletion allele uu3731, designated ΔJRS1) was selected for further analysis. These F1 *nkx3.2*$^{+/\Delta JRS1}$ fish were incrossed to produce F2 *nkx3.2*$^{\Delta JRS1/\Delta JRS1}$ fish for analysis. JRS1 deletion was confirmed by Sanger sequencing (Eurofins Genomics) using the same PCR primers.

## Skeletal staining

Zebrafish larvae and juveniles were stained with Alcian blue and alizarin red following a protocol modified from *Walker and Kimmel, 2007*, previously described by *Waldmann et al., 2021*. Specimens were imaged using a Leica M205 FCA fluorescence stereomicroscope with attached Leica DFA 7000T camera.

## Optical projection tomography

A custom-built OPT (*Sharpe et al., 2002*; *Zhang et al., 2020*) system was used for imaging of 9 dpf skeletal stained zebrafish larvae. The OPT system, reconstruction algorithms, and alignment workflow were based on the previously described method (*Allalou et al., 2017*). All larvae were kept in 99% glycerol before being loaded into the system for imaging. The rotational images were acquired using a ×3 telecentric objective with a pixel resolution of 1.15 µm/pixel. The tomographic 3D reconstruction was done using a filtered back projection algorithm in MATLAB (Release R2015b; MathWorks, Natick, MA) together with the ASTRA Toolbox (*Palenstijn et al., 2013*). For the data alignment, the registration toolbox elastix (*Klein et al., 2010*; *Shamonin et al., 2013*) was used. To reduce the computational time all 3D volumes in the registration were down-sampled to half the resolution.

The registration workflow was similar to the methods described by *Allalou et al., 2017* where the wild-type fish were initially aligned and used to create an average reference fish using an Iterative Shape Averaging (ISA) algorithm (*Rohlfing et al., 2001*). All wild-type (*N* = 12), *nkx3.2*$^{+/\Delta JRS1}$ (*N* = 10), and *nkx3.2*$^{\Delta JRS1/\Delta JRS1}$ (*N* = 8) zebrafish were then aligned to the reference.

## Whole-mount in situ hybridization

Primers were designed for the zebrafish *nkx3.2* gene sequence forward 5′-CTTCAACCACCAGCGTT ATCTC-3′ and reverse 5′-ACATGTCTAGTAAACGGGCGA-3′. Fragments were cloned from zebrafish

cDNA into the pCR II TOPO vector and antisense RNA probes were synthesized with either SP6 or T7 RNA polymerase and digoxigenin labelling mix (Roche). In situ hybridization on zebrafish whole-mount 48 and 56 hpf embryos was performed as previously described (*Filipek-Górniok et al., 2013*). Genotyping, imaging, and quantification of in situ stained zebrafish embryos were performed according to the previously published protocol by *Dobrzycki et al., 2020b*. Specimens were imaged using a Leica M205 FCA fluorescence stereomicroscope with attached Leica DFA 7000T camera. Normal distribution of obtained pixel intensity values (relative to the unstained tissue background) assigned to the genotypes was verified with a Kolmogorov–Smirnov test. The differences in wild-type and $nkx3.2^{+/\Delta JRS1}$ ($N$ = 16 at 48 hpf, $N$ = 18 at 56 hpf) versus $nkx3.2^{\Delta JRS1/\Delta JRS1}$ ($N$ = 17 at 48 hpf, $N$ = 18 at 56 hpf) groups was analysed with unpaired *t*-tests.

## Quantitative PCR analysis of gene expression

6 dpf larvae produced from incrossing heterozygous parents were euthanized with an overdose of MS-222 (300 mg/l) and the tails were removed for genotyping while the head and remaining body were stored in RNALater (Invitrogen). Total RNA was extracted with Trizol reagent (Fisher) from a pool of seven larvae of each genotype, with three biological replicates per genotype. To prevent genomic DNA contamination, extractions were DNase-treated using the Turbo DNA-free kit (Ambion). cDNA was synthesized from 200 ng total RNA from each sample using the SuperScript IV First-Strand cDNA Synthesis kit (Invitrogen) with random hexamers in a 20 µl reaction. qPCR was performed on all samples in technical triplicates with PowerUp SYBR Green Master Mix using a 7500 Real Time PCR System (Applied Biosystems) and the following primers: *rpl13a* forward 5'-TCTGGAGGACTGTAAGAGGTATGC-3'; *rpl13a* reverse 5'-AGACGCACAATCTTGAGAGCAG-3'; *nkx3.2* forward 5'-ACTGCGTGTCCGACTGTAACAC-3'; *nkx3.2* reverse 5'-GTCTCGGTGAGTTTGAGGGA-3'. Amplicon sizes for these target genes were 148 and 187 bp, respectively. A dissociation step was performed at the end of the analysis to verify the specificity of the products, and standard curves were generated from pooled cDNA from all samples in triplicate for each target gene to verify the efficiency of the primers ($R^2$ > 0.99). *nkx3.2* expression was normalized to *rpl13a* levels and relative quantification of gene expression was calculated using the *Pfaffl, 2001* method, displaying the fold difference in heterozygous and homozygous mutants relative to wild type, which was set to 1.0. FDR-adjusted p values (*Benjamini and Hochberg, 1995*) are reported from pairwise Wilcoxon tests.

## Geometric morphometric analysis

We employed 2D geometric morphometric analysis using 100 landmarks defined along the edge of Meckel's cartilage at the jaw joint interface. From maximum projection images generated by OPT, standardized with a lateral orientation, the shapes of Meckel's cartilage were traced in Adobe Illustrator 2020. These shapes were then aligned by orientation and the joint interfacing surface, but not resized, before most of the anterior portion of the shapes were cut away to a standardized extent, leaving shapes representing just the posterior head of Meckel's cartilage. These shapes were digitized using 100 equidistant landmarks using tpsDig2 (v2.3) (*Rohlf, 2017*), and imported into R for analysis (*R Development Core Team, 2021*). The two landmarks at the ends of each curve were used as fixed landmarks, and the remaining 98 as semi-landmarks.

Generalized procrustes analysis was used to align coordinates of the landmarks for subsequent morphospace analysis using the geomorph package (v3.3.2) (*Adams et al., 2021*) and bivariate plots of the PC axis were generated using the borealis package (*Angelini, 2021*). FDR-adjusted p values (*Benjamini and Hochberg, 1995*) are reported from pairwise comparisons of group means after accounting for allometric differences.

## Acknowledgements

We thank Professor Per E Ahlberg, Dr Erika Kague, and Dr Sophie Sanchez for providing thoughtful comments on the manuscript. Professor Per E Ahlberg also covered some lab expenses. We thank Professor Masataka Okabe at the Jikei University School of Medicine for sharing the early genomic scaffold of *Polypterus senegalus*, Dr Qingming Qu for *P. senegalus* gDNA, Dr Christine Boisvert for *Callorhinchus milii* gDNA, Dr Henning Onsbring Gustafson for help with initial cloning of the zebrafish construct, Dr Mohamad Bazzi for help with geometric morphometrics, Dr Emmanouil Tsakoumis for help with RNA extraction and qPCR, and Philipp Pottmeier for access to and assistance with the 7500

Real Time PCR System. We also thank the following Erasmus students for their help during various stages of the project: Onur Özer at Bilkent University, Thibaut D'Hooge and Branco Vanhaverbeke at University College Ghent. Finally, we thank the editor and three external reviewers for their critical comments and suggestions that strengthened the final version of our manuscript. TH was supported by Vetenskapsrådet (Starting grant 621-2012-4673). The development of the OPT system was funded by a development project at SciLifeLab Uppsala (2017) and a Technology Development grant at SciLifeLab (2018), both awarded to AA.

## Additional information

### Funding

| Funder | Grant reference number | Author |
|---|---|---|
| Vetenskapsrådet | Starting Grant 621-2012-4673 | Tatjana Haitina |
| Science for Life Laboratory | Development Project 2017 | Amin Allalou |
| Science for Life Laboratory | Technology Development grant 2018 | Amin Allalou |

The funders had no role in study design, data collection, and interpretation, or the decision to submit the work for publication.

### Author contributions

Jake Leyhr, Data curation, Formal analysis, Validation, Investigation, Visualization, Methodology, Writing – original draft, Writing - review and editing; Laura Waldmann, Formal analysis, Validation, Investigation, Visualization, Methodology, Writing – original draft; Beata Filipek-Górniok, Data curation, Formal analysis, Methodology, Writing – original draft; Hanqing Zhang, Software, Formal analysis, Visualization, Methodology; Amin Allalou, Software, Formal analysis, Funding acquisition, Visualization, Methodology, Writing – original draft; Tatjana Haitina, Conceptualization, Data curation, Supervision, Funding acquisition, Validation, Investigation, Visualization, Writing – original draft, Project administration, Writing - review and editing

### Author ORCIDs

Jake Leyhr http://orcid.org/0000-0003-1815-7818
Tatjana Haitina http://orcid.org/0000-0002-8754-5534

### Ethics

All animal experimental procedures were approved by the local ethics committee for animal research in Uppsala, Sweden (permit number C161/4 and 5.8.18-18096/2019). All procedures for the experiments were performed in accordance with the animal welfare guidelines of the Swedish National Board for Laboratory Animals.

### Decision letter and Author response

Decision letter https://doi.org/10.7554/eLife.75749.sa1
Author response https://doi.org/10.7554/eLife.75749.sa2

## Additional files

### Supplementary files

• Transparent reporting form

### Data availability

OPT data raw image stacks used to generate Figure 7B–G and Figure 7—figure supplement 1 are deposited in Dryad. Figure 2_figure supplement 2_Source Data 1, Figure 3-Source Data 1, Figure 6_Source Data 1, Figure 6_Source Data 2, Figure 7_Source Data 1, and Figure 7_Source Data 2 contain the numerical data used to generate the figures.

The following dataset was generated:

| Author(s) | Year | Dataset title | Dataset URL | Database and Identifier |
|---|---|---|---|---|
| Leyhr J, Waldmann L, Filipek-Górniok B, Zhang H, Allalou A, Haitina T | 2022 | OPT data from: A novel cis-regulatory element drives early expression of Nkx3.2 in the gnathostome primary jaw joint | https://dx.doi.org/10.5061/dryad.p5hqbzkqw | Dryad Digital Repository, 10.5061/dryad.p5hqbzkqw |

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
