## [Editor Report]

In this elegant and important study, Leyhr et al. identify the first potent nkx3.2 jaw joint enhancer, which they show to be deeply conserved across gnathostomes and absent from jawless fishes. The data are convincing and beautifully presented, supporting the hypothesis that this enhancer arose with the origin of hinged jaws during vertebrate evolution and is required for some aspects of early joint development in zebrafish. The work has important implications both for our basic understanding of enhancer function and evolution as well as potential genetic causes of craniofacial defects in humans.

---

## [Decision Letter]

**Decision letter after peer review:**

Thank you for submitting your article "An evolutionarily conserved cis-regulatory element of *Nkx3.2* contributes to early jaw joint morphology in zebrafish" for consideration by *eLife*. Your article has been reviewed by 3 peer reviewers, and the evaluation has been overseen by a Reviewing Editor and Marianne Bronner as the Senior Editor. The following individuals involved in review of your submission have agreed to reveal their identity: Chrissy Hammond (Reviewer #1); Thomas F Schilling (Reviewer #3).

Essential revisions:

(1) Determine changes in nkx3.2 expression in the enhancer-less mutants, including characterisation of any tissue-specific expression changes using in situ hybridisation.

(2) Check the available lamprey genome assemblies for presence of a JRS-like sequence. This is essential before making strong conclusions about its absence in agnathans.

(3) Validate one or more of the candidate transcription factors binding to JRS1 using ChIP, or, if ChIP-type tools are not available, move the speculative section on transcription factor binding sites and associated literature on their function from the Results to the Discussion.

The full reviews are appended below for your information.

*Reviewer #1 (Recommendations for the authors):*

The work throughout is well performed and the imaging is beautiful. The main points are well made and well evidenced.

The section on matching of the JRS1 to transcription factors doesn't flow as well as the rest of the manuscript. It would have been nice to see validation of that one or more of the TFs discussed binding to the JRS using CHiP.

As the authors introduce the idea of OA (in relation to the factors that bind the JRS1)and as nkx3.2 is known to inhibit chondrocyte maturation in addition to the Alcian blue/AR and joint shape ideally one might show maturation state of chondrocytes using progression from *Sox9*/10 positivity to Col2+ and potentially to Col10 in the fish in which the JRS1 is deleted. While there are transgenics for these, there are also decent antibodies that work in zebrafish (see for example https://faseb.onlinelibrary.wiley.com/doi/full/10.1096/fj.202101167R) which would allow relatively rapid assessment.

*Reviewer #2 (Recommendations for the authors):*

While this study is clear and well designed, it could be meaningfully strengthened by making several modifications to the text and performing a few additional analyses, none of which would require additional models to be generated.

1. The authors note that they were unable to detect the JRS1 sequence in the hagfish genome, but do not comment about lamprey. This is worth checking, as there are several builds of the lamprey genome available. I can find an annotated "nkx3-2" gene in the SIMRbase genome (https://simrbase.stowers.org/sealamprey) though nothing very promising on the UCSC genome browser.

2. The authors should further probe and discuss the role of Hox genes in regulating Nkx3.2 expression. Notably, a previous study in zebrafish established that nkx3.2 is suppressed in the second pharyngeal arch by Hox activity (Miller et al. 2004). Specifically, moz mutants lacking Hox expression in this arch and hoxa2b/b2a morphants both showed ectopic second arch nkx3.2 expression. I was thus intrigued to note that the authors identified a predicted Meis2 – though not a Hox – binding motif in the second conserved motif of the JRS1 enhancer (Figure 2C). Meis proteins are of course known for being key Hox cofactors, along with Pbx proteins. To check whether a stronger connection between the JRS1 enhancer and Hox regulation could be made using existing data, I quickly consulted ChIP-seq datasets for Hoxa2, Meis, and Pbx in mouse pharyngeal arch tissue that were published by Donaldson et al. (2012) and Amin et al. (2015). Distinct peaks and summits for all three (Hoxa2, Pbx, and Meis) were present directly over the mouse JRS1 sequence. This suggests that the same element that activates nkx3.2 expression in the first arch may be directly repressed in the second arch by a Hox-Pbx-Meis complex.

3. This notion may also be relevant for the authors' discussion of the chondrichthyan element (lines 468-476) – is the Meis motif intact in the elephant shark sequence?

4. It is intriguing that the JRS1 enhancer was not detected in Acanthopterygii species. The authors note that it may have been translocated to a different genomic location rather than completely lost – to test this, they should perform unbiased Ensembl BLAST searches of these fish genomes using, e.g., the spotted gar sequence as bait, outside the original locus. If they do not find any evidence in favor of translocation, they could comment on whether there are any potential conserved shadow enhancers that could have taken over regulation of nkx3.2 expression in these species. I appreciate that they note that fish in this group show unusual upper jaw protrusability, which may be a consequence of altered selective pressures on the jaw joint.

5. The authors detected a conserved Hey1 binding motif in the JRS1 enhancer and noted that hey1 was reported to be expressed downstream of Notch activity in the dorsal domain of the zebrafish pharyngeal arches (Zuniga et al. 2010). They should rephrase this comment to note that those previous authors did indeed find that nkx3.2 expression was dorsally expanded in Notch loss-of-function mutants, supporting the notion that Notch activity, possibly via Hey1, is suppressing its expression dorsally.

6. The finding that JRS1:mCherry expression was still expressed in the fused jaw joint of nkx3.2-/- fish argues against Nkx3.2 playing a significant role in autoregulation of its own expression through the JRS1 enhancer (as proposed in lines 176-177). They should consider noting this in the discussion.

7. Deletion of the enhancer caused a mild phenotype that resolved with age. This finding, together with the fact that they saw no difference in nkx3.2 transcript levels at 6 dpf, suggests that shadow enhancers may activate later and compensate for early loss of expression. However, the qPCR experiment appears to have been done on whole larvae, meaning that normal levels of nkx3.2 in other parts of the body could have masked an effect specific to the jaw joint. To resolve this, the authors should assess nkx3.2 expression in the jaw region of the JRS1 mutants by in situ hybridization, which, although not quantitative, is sensitive enough to reveal differences in local expression levels. In addition (or alternatively), the authors should perform the qPCR at an earlier stage of development to test for diminished expression at the stage when the joint is first being specified. Tail biopsy-based genotyping of zebrafish larvae can be done as early as 24 hpf.

*Reviewer #3 (Recommendations for the authors):*

(1) Look further into changes in Nkx3.2 expression with deletion of the JRS in zebrafish. While there appear to be no statistically significant changes at 6 days postfertilization (6 dpf) homozygotes show slightly reduced expression. Has this experiment been done at 2-5 dpf when jaw joint development is initiated? If so and no changes were detected this might suggest that either loss of the JRS disrupts spatial patterns of Nkx3.2 expression, which the authors should test with in situ hybridization, or the enhancer may act on other nearby genes.

(2) Quantify variation in the jaw joint phenotype in the homozygous JRS mutants.

(3) Check the available lamprey genomes for presence of a JRS-like sequence. If absent this would bolster the argument for an involvement of the JRS in the origin of jaws during vertebrate evolution.

(4) Move the long list of transcription factor binding sites in the JRS and literature describing their functions that is currently included in the Results into the Discussion.

---

## [Author Response]

Essential revisions:(1) Determine changes in nkx3.2 expression in the enhancer-less mutants, including characterisation of any tissue-specific expression changes using in situ hybridisation.

We have added an analysis of jaw joint-specific *nkx3.2* expression changes at 48 and 56 hpf to the manuscript, demonstrating that the loss of JRS1 severely reduces the early expression of *nkx3.2* (lines 471-473, 795-804, 1080-1097, Figure 6K-O)

(2) Check the available lamprey genome assemblies for presence of a JRS-like sequence. This is essential before making strong conclusions about its absence in agnathans.

We have repeated our searches for a JRS1-like sequence in both hagfish and the sea lamprey genomes and can now make more strong claims about its absence in agnathans (lines 123-146, 161-163, 882-884, Figure 1, Figure 1A).

(3) Validate one or more of the candidate transcription factors binding to JRS1 using ChIP, or, if ChIP-type tools are not available, move the speculative section on transcription factor binding sites and associated literature on their function from the Results to the Discussion.

We have been able to validate Meis binding at JRS1 by accessing the publicly available datasets generated as part of two previous publications, in addition to moving the detailed description of all of the predicted binding sites and their associated literature to the Discussion section as suggested (lines 185-192, 626-705, Figure 2 —figure supplement 2).

The full reviews are appended below for your information.Reviewer #1 (Recommendations for the authors):The work throughout is well performed and the imaging is beautiful. The main points are well made and well evidenced.

We thank the reviewer for the comments.

The section on matching of the JRS1 to transcription factors doesn't flow as well as the rest of the manuscript. It would have been nice to see validation of that one or more of the TFs discussed binding to the JRS using CHiP.

We have moved the text describing the predicted transcription factor binding sites to the discussion, and providing a briefer description in the Results section to improve the flow. Reviewer 3 directed us to existing ChIP-seq datasets in E11.5 mouse embryonic branchial arches showing the strong presence of Meis bound to JRS1, validating the predicted Meis2 binding. This validation is described in the Results section (lines 185-192) and has been added to the discussion (lines 626-655).

As the authors introduce the idea of OA (in relation to the factors that bind the JRS1)and as nkx3.2 is known to inhibit chondrocyte maturation in addition to the Alcian blue/AR and joint shape ideally one might show maturation state of chondrocytes using progression from Sox9/10 positivity to Col2+ and potentially to Col10 in the fish in which the JRS1 is deleted. While there are transgenics for these, there are also decent antibodies that work in zebrafish (see for example https://faseb.onlinelibrary.wiley.com/doi/full/10.1096/fj.202101167R) which would allow relatively rapid assessment.

We thank the reviewer for raising this suggestion and we agree that it would certainly be interesting to determine how the chondrocyte maturation state relates to the observed phenotypes, but feel this goes beyond the scope of the current paper which is primarily aimed at characterising JRS1 and demonstrating that it influences *nkx3.2* expression with phenotypic effects. Future studies may address precisely how different *nkx3.2* expression levels influence jaw joint morphology.

Reviewer #2 (Recommendations for the authors):While this study is clear and well designed, it could be meaningfully strengthened by making several modifications to the text and performing a few additional analyses, none of which would require additional models to be generated.1. The authors note that they were unable to detect the JRS1 sequence in the hagfish genome, but do not comment about lamprey. This is worth checking, as there are several builds of the lamprey genome available. I can find an annotated "nkx3-2" gene in the SIMRbase genome (https://simrbase.stowers.org/sealamprey) though nothing very promising on the UCSC genome browser.

We agree, and thank the reviewer for raising this point. To be as rigorous as possible, we have repeated the synteny analysis of both the hagfish and sea lamprey genomes to search for *Nkx3.2* gene loci, and identified in lamprey a *Nkx3.2* locus with some conservation in synteny to hagfish and the gnathostomes, in addition to an *Nkx3.2-*like gene in both lamprey and hagfish with different syntenic surroundings. We performed the mVISTA analysis on all four *Nkx3*.2/*Nkx3.2-*like loci from hagfish and lamprey, and were unable to identify any JRS1-like sequences proximal to any of them. This analysis is described in the updated Results section (lines 123-146 and 161-163), and we have updated Figure 1 and Figure 2A to show the results from the syntenic *Nkx3.2* loci only. We decided not to focus on the nonsyntenic *Nkx3.2*-like loci as they likely represent Nkx-family paralogues, which are challenging to robustly annotate or describe in the absence of a comprehensive study of the diversity of the gene family in vertebrates.

2. The authors should further probe and discuss the role of Hox genes in regulating Nkx3.2 expression. Notably, a previous study in zebrafish established that nkx3.2 is suppressed in the second pharyngeal arch by Hox activity (Miller et al. 2004). Specifically, moz mutants lacking Hox expression in this arch and hoxa2b/b2a morphants both showed ectopic second arch nkx3.2 expression. I was thus intrigued to note that the authors identified a predicted Meis2 – though not a Hox – binding motif in the second conserved motif of the JRS1 enhancer (Figure 2C). Meis proteins are of course known for being key Hox cofactors, along with Pbx proteins. To check whether a stronger connection between the JRS1 enhancer and Hox regulation could be made using existing data, I quickly consulted ChIP-seq datasets for Hoxa2, Meis, and Pbx in mouse pharyngeal arch tissue that were published by Donaldson et al. (2012) and Amin et al. (2015). Distinct peaks and summits for all three (Hoxa2, Pbx, and Meis) were present directly over the mouse JRS1 sequence. This suggests that the same element that activates nkx3.2 expression in the first arch may be directly repressed in the second arch by a Hox-Pbx-Meis complex.

We thank the reviewer for this suggestion and for checking the datasets themselves for the relevant binding. We have added a summary of the Meis/Hoxa2/Pbx binding data from Donaldson et al. (2012) and Amin et al. (2015) as new figure supplement: Figure 2 —figure supplement 2. These data and their implications are now described in lines 185-192 of the results and in lines 626-655 of the discussion.

3. This notion may also be relevant for the authors' discussion of the chondrichthyan element (lines 468-476) – is the Meis motif intact in the elephant shark sequence?

The predicted Meis2 binding motif is indeed intact in the elephant shark sequence (and in other chondrichthyans), and we have added this fact to the discussion (lines 738-743).

4. It is intriguing that the JRS1 enhancer was not detected in Acanthopterygii species. The authors note that it may have been translocated to a different genomic location rather than completely lost – to test this, they should perform unbiased Ensembl BLAST searches of these fish genomes using, e.g., the spotted gar sequence as bait, outside the original locus. If they do not find any evidence in favor of translocation, they could comment on whether there are any potential conserved shadow enhancers that could have taken over regulation of nkx3.2 expression in these species. I appreciate that they note that fish in this group show unusual upper jaw protrusability, which may be a consequence of altered selective pressures on the jaw joint.

We agree that it would be useful to more rigorously test this conjecture, and we have now performed the suggested BLAST searches. We did this using both the spotted gar and bichir JRS1 sequences as queries against the whole genomes of the teleost species, including additional species from a more complete sampling of clades near the origin of Acanthopterygii. The methods section was updated accordingly, and the results are described in lines 223-250. We supported the text with a new Figure 3 depicting both the BLASTN search results (E-values and locations of top BLASTN hits in each genome) and the new summary of how JRS1 seems to have been lost. In the first version of the manuscript we proposed that JRS1 had been lost or translocated at the base of Acanthopterygii, but now with the inclusion of additional species and a more robust analysis, we show that JRS1 began to be partially lost/diverged after the divergence of Trachichthyiformes with the remainder of the Acanthopterygian clades, and that JRS1 is only completely lost (or unable to be identified by the current analysis) near the base of Percomorpha, after Ophidiiformes diverged. The Discussion section has been updated accordingly (lines 745-771).

We added a comment on the presence of additional shadow enhancers in lines 771-775.

5. The authors detected a conserved Hey1 binding motif in the JRS1 enhancer and noted that hey1 was reported to be expressed downstream of Notch activity in the dorsal domain of the zebrafish pharyngeal arches (Zuniga et al. 2010). They should rephrase this comment to note that those previous authors did indeed find that nkx3.2 expression was dorsally expanded in Notch loss-of-function mutants, supporting the notion that Notch activity, possibly via Hey1, is suppressing its expression dorsally.

We have rephrased this comment to add this important note (lines 659-666).

6. The finding that JRS1:mCherry expression was still expressed in the fused jaw joint of nkx3.2-/- fish argues against Nkx3.2 playing a significant role in autoregulation of its own expression through the JRS1 enhancer (as proposed in lines 176-177). They should consider noting this in the discussion.

We agree, and have added this caveat (lines 700-705).

7. Deletion of the enhancer caused a mild phenotype that resolved with age. This finding, together with the fact that they saw no difference in nkx3.2 transcript levels at 6 dpf, suggests that shadow enhancers may activate later and compensate for early loss of expression. However, the qPCR experiment appears to have been done on whole larvae, meaning that normal levels of nkx3.2 in other parts of the body could have masked an effect specific to the jaw joint. To resolve this, the authors should assess nkx3.2 expression in the jaw region of the JRS1 mutants by in situ hybridization, which, although not quantitative, is sensitive enough to reveal differences in local expression levels. In addition (or alternatively), the authors should perform the qPCR at an earlier stage of development to test for diminished expression at the stage when the joint is first being specified. Tail biopsy-based genotyping of zebrafish larvae can be done as early as 24 hpf.

We agree that further quantification of *nkx3.2* gene expression differences is required to support our conclusions. To this end, we performed in situ hybridisation to assess *nkx3.2* gene expression in the jaw joints of 48 and 56 hpf embryos. Following the protocol described by Dobrzycki et al. (2020), we were able to quantify the intensity of ISH staining in the jaw joints and show that homozygous JRS1 deletion mutants displayed a ~45% reduction in staining intensity, demonstrating that *nkx3.2* expression was severely reduced in the absence of JRS1. These results have been added to Figure 6 in panels K-O, described in lines 471-473, and discussed in lines 795-804. The methods section at lines 1080-1097 has also be updated accordingly. We are still unable to distinguish whether the absence of *nkx3.2* expression differences between genotypes at 6 dpf is the result of other enhancers “taking over”, or that extra-jaw-joint expression swamps the signal, and have added this caveat clearly at lines 815-822.

Reviewer #3 (Recommendations for the authors):(1) Look further into changes in Nkx3.2 expression with deletion of the JRS in zebrafish. While there appear to be no statistically significant changes at 6 days postfertilization (6 dpf) homozygotes show slightly reduced expression. Has this experiment been done at 2-5 dpf when jaw joint development is initiated? If so and no changes were detected this might suggest that either loss of the JRS disrupts spatial patterns of Nkx3.2 expression, which the authors should test with in situ hybridization, or the enhancer may act on other nearby genes.

We agree that further quantification of *nkx3.2* gene expression differences is required to support our conclusions. To this end, we performed in situ hybridisation to assess nkx3.2 gene expression in the jaw joints of 48 and 56 hpf embryos. Following the protocol described by Dobrzycki et al. (2020a), we were able to quantify the intensity of ISH staining in the jaw joints and show that homozygous JRS1 deletion mutants displayed a ~45% reduction in staining intensity, demonstrating that *nkx3.2* expression was severely reduced in the absence of JRS1. These results have been added to Figure 6 in panels K-O, described in lines 471-473, and discussed in lines 795-804. The methods section at lines 1080-1097 has also be updated accordingly. We are still unable to distinguish whether the absence of *nkx3.2* expression differences between genotypes at 6 dpf is the result of other enhancers “taking over”, or that extra-jaw-joint expression swamps the signal, and have added this caveat clearly at lines 815-822.

(2) Quantify variation in the jaw joint phenotype in the homozygous JRS mutants.

The most rigorous presentation of the variation in jaw joint phenotype is present in Figure 7A, but we have added numbers to Figure 6C, D, F, G, I, J to quantify how representative the images are of the mutant phenotypes.

(3) Check the available lamprey genomes for presence of a JRS-like sequence. If absent this would bolster the argument for an involvement of the JRS in the origin of jaws during vertebrate evolution.

We agree, and thank the reviewer for raising this point. To be as rigorous as possible, we have repeated the synteny analysis of both the hagfish and sea lamprey genomes to search for *Nkx3.2* gene loci, and identified in lamprey a *Nkx3.2* locus with some conservation in synteny to hagfish and the gnathostomes, in addition to an *Nkx3.2-*like gene in both lamprey and hagfish with different syntenic surroundings. We performed the mVISTA analysis on all 4 *Nkx3*.2/*Nkx3.2-*like loci from hagfish and lamprey, and were unable to identify any JRS1like sequences proximal to any of them. This analysis is described in the updated Results section (lines 123-146 and 161-163), and we have updated Figure 1 and Figure 2A to show the results from the syntenic *Nkx3.2* loci only. We decided not to focus on the non-syntenic *Nkx3.2*-like loci as they likely represent Nkx-family paralogues which are challenging to robustly annotate or describe in the absence of a comprehensive study of the diversity of the gene family in vertebrates.

(4) Move the long list of transcription factor binding sites in the JRS and literature describing their functions that is currently included in the Results into the Discussion.

We have moved this text into the discussion.